**Evaluation of the WRF-Chem Performance for the air pollutants over the United Arab Emirates**

**Yesobu Yarragunta[1], Diana Francis[1*], Ricardo Fonseca[1] and Narendra Nelli[1]**

[1] Environmental and Geophysical Sciences (ENGEOS) Lab, Khalifa University of Science and Technology, P. O. Box 127788, Abu Dhabi, United Arab Emirates

*Correspondence to*: Diana Francis (diana.francis@ku.ac.ae)

## Abstract

This study presents a comprehensive evaluation of the Weather Research and Forecasting model coupled with Chemistry (WRF-Chem) in simulating meteorological parameters and concentrations of air pollutants across the United Arab Emirates (UAE) for June and December 2022, representing the contrasting summer and winter climatic conditions. The assessment of WRF-Chem performance involves comparisons with ground-based observations for meteorological parameters and satellite retrievals from the TROPOspheric Monitoring Instrument (TROPOMI) for gaseous pollutants and the Moderate Resolution Imaging Spectroradiometer (MODIS) for aerosols. The comparison with TROPOMI column concentrations demonstrates that WRF-Chem performs well in simulating the spatio-temporal patterns of total column CO and tropospheric column $NO_2$, $O_3$, despite certain deficiencies in modeling tropospheric $NO_2$ column concentrations. In particular, WRF-Chem shows a strong correlation with TROPOMI retrievals, with correlation coefficients ranging from 0.53 to 0.82 during summer and 0.40 to 0.69 during winter for these gaseous pollutants. The model tends to overestimate $NO_2$ levels, with a higher discrepancy observed in summer ($0.50 \times 10^{15}$ molecules/cm²) compared to winter ($0.18 \times 10^{15}$ molecules/cm²). In comparison with TROPOMI-CO data, the discrepancies are more pronounced in winter, with an underestimation of $0.12 \times 10^{18}$ molecules/cm². Additionally, WRF-Chem consistently overestimates ozone levels in both seasons. WRF-Chem also exhibits a moderate correlation with both AERONET and MODIS AOD measurements. The correlation at Mezaira is 0.60, while a correlation of 0.65 is observed with MODIS AOD. However, the model tends to overestimate AOD, with a bias of 0.46 at Mezaira and 0.35 compared to MODIS AOD.

Meteorological evaluations reveal that the model generally overestimated T2m in summer (≤0.2°C) and underestimated it in winter (~3°C) with correlation coefficients between 0.7 and 0.85. Temperature biases are linked to surface property representation and model physics. For WS10m, biases were within ±0.5 m/s, indicating good agreement, although overestimations suggest deficiencies in surface drag parameterization. The dry bias observed was consistent with other studies due to dry soil, inaccurate mesoscale circulation representation, and bias in forcing data. The model also overestimated incoming shortwave radiation by ~30 W/m² in December due to reduced cloud cover. Night-time cold and dry biases were observed due to more substantial wind speeds and cooler air advection. Comparisons with ERA5 reanalysis showed regional T2m variations with high correlation coefficients (0.97 in summer, 0.92 in winter). Both WRF-Chem and ERA5 displayed consistent seasonal patterns in the planetary boundary layer, correlating with temperature changes and indicating good overall model performance.

**Keywords:** Air quality modeling, air pollutants, TROPOMI satellite retrievals, MODIS, WRF-Chem, UAE.

**Key points:**
- First high-resolution WRF-Chem air quality modeling study over the United Arab Emirates (UAE)
- WRF-Chem's ability to simulate meteorological parameters and pollutant levels over the UAE is assessed during summer and winter in 2022.
- The model strongly correlated with TROPOMI satellite data, achieving correlation coefficients of 0.53-0.82 in summer and 0.40-0.69 in winter for gaseous pollutants.
- Lower model skill in simulating tropospheric $NO_2$ columns, in contrast to the more accurate modeling of total CO and tropospheric $O_3$ columns, particularly in summer.
- WRF-Chem demonstrated a moderate correlation with AERONET and MODIS for AOD during the summer, with correlation coefficients of 0.60 and 0.65, respectively.
- Meteorological analysis revealed a tendency to overestimate surface temperature by 0.2 °C in summer and underestimate it by 3 °C in winter across land regions. Surface wind speed is overestimated by 0.1-0.5 m/s in both seasons across various regimes.

## 1. Introduction

The United Arab Emirates (UAE), a federation of seven emirates, has undergone rapid urbanization and industrialization over the last five decades, which has had a profound impact on its air quality (Ramadan, 2015). The major factors affecting air quality in the UAE include emissions from industrial activities, vehicular traffic, construction projects (Teixido et al., 2021), and occasionally, natural phenomena such as dust storms, which are quite prevalent in the region due to its desert climate (Environment Agency – Abu Dhabi, 2018; Francis et al., 2020; 2022b; Karagulian et al., 2019). The rapid economic growth of the UAE, especially in cities like Dubai and Abu Dhabi, has led to a surge in energy demand and water, the latter obtained from desalination and cloud seeding activities (Wehbe et al., 2023), largely met through the burning of fossil fuels (Shahbaz et al., 2014). This has resulted in increased emissions of pollutants like oxides of nitrogen (NOx), sulfur dioxide ($SO_2$), particulate matter (PM), and volatile organic compounds (VOCs). Moreover, the heavy traffic in urban areas contributes to the elevated levels of ground-level ozone and particulate pollution (Abuelgasim & Farahat, 2020; Li et al., 2010). Understanding the dynamics of air quality in the UAE involves considering both the environmental challenges posed by rapid development and the steps being taken to mitigate these impacts. The pursuit of balancing economic growth with environmental sustainability is central to this discourse. This area of study is not only vital for ensuring the health and well-being of the population but also plays a crucial role in the UAE's vision for a sustainable future.

The swift urban expansion in the UAE, which is expected to continue in the coming decades, could intensify air pollution sources. With surface observations sparse in this region, satellite remote sensing becomes a crucial method for air quality monitoring (Chudnovsky et al., 2014; Fonseca et al., 2023; Francis et al., 2023). What is more, satellite measurements themselves fall short in clarifying the different atmospheric processes responsible for peak pollution levels. Consequently, integrating chemistry transport models with satellite-derived and ground-based observations can significantly improve our understanding of pollutant emissions, distribution, transport, and transformation in the targeted regions (Eltahan et al., 2018; Li et al., 2018; Yarragunta et al., 2020; Yin et al., 2021). Air quality (AQ) modelling is dedicated to unravelling the complicated aspects of atmospheric chemistry and transport across both global and regional levels, as explored in numerous studies conducted around the world (Emmons et al., 2010; Kumar et al., 2011, 2018; Tie et al., 2001; Yarragunta et al., 2019, 2020, 2021).

Despite facing limitations due to the often low spatial and temporal resolution of observational data, AQ models effectively generate detailed air quality information for remote regions (e.g., Guo et al., 2024a). They predict the formation and removal of air pollutants and facilitate a thorough examination of the transport and photo-chemical transformation of trace gases following their emission into the atmosphere (Archer-Nicholls et al., 2015; Georgiou et al., 2018; Nhu et al., 2021; Sicard et al., 2021). They are also employed globally for operational air quality forecasting (Jena et al., 2021; Koo et al., 2012; Kumar et al., 2012, 2021; Srinivas et al., 2016; Zhang et al., 2012). Air quality models are categorized into two types: 'fully coupled' models, which integrate interactions between chemistry and meteorology, and 'offline' models, where chemistry and meteorology simulations are conducted independently (Gao & Zhou, 2024). Some of  state of the art AQ models include the Weather Research and Forecasting (WRF) model coupled with chemistry (WRF-Chem;  Grell et al., 2005; Skamarock et al., 2008), WRF-Chem-MADRID (Model of Aerosol Dynamics, Reaction, Ionization and Dissolution; Zhang et al., 2010), CESM2 (Community Earth System Model version 2; Emmons et al., 2020), CHIMERE (Menut et al., 2021), LOTOS-EUROS(v2.0) (Long Term Ozone Simulation European Operational Smog; Manders et al., 2017) and COSMO/MESSy (Consortium for Small Scale Modelling/ Modular Earth Submodel System; Kerkweg & Jöckel, 2012). However, before using these AQ models for operational or research applications, it is crucial to conduct thorough evaluations to assess the quality of their predictions. The AQ model chosen for the current study is WRF-Chem with its foundational meteorological component, WRF. WRF-Chem has been used for research studies in the Arabian Peninsula (Parajuli et al. 2019, 2023, 2024), with the meteorological component optimized for simulations over the region (Chaouch et al., 2017; Nelli et al., 2020; Abida et al., 2022; Fonseca et al. 2020, 2021, 2022a).

The majority of studies conducted in the UAE and similar arid regions have primarily focused on evaluation of meteorological parameter including temperature, humidity, wind, and solar radiation (Parajuli et al., 2019; Nelli et al., 2020; Fonseca et al., 2020, 2021) with a few others investigating the particulate matter (PM) dynamics, especially mineral dust. For instance, Ukhov et al., (2021) noted inaccuracies in the WRF-Chem model related to the commonly used bulk Goddard Chemistry Aerosol Radiation and Transport (GOCART; Chin et al., 2022) aerosol module, affecting $PM_{2.5}$ and $PM_{10}$ diagnostics. Karagulian et al. (2019) highlighted the effectiveness of integrating WRF-Chem model simulations with satellite and ground observations to understand and predict the impact of severe dust storms on air quality.

Karumuri et al., (2022) reported significant air quality changes due to COVID-19 lockdown
measures, with reduced trace gas concentrations but increased particulate matter from dust
activities, the latter stressed by Francis et al. (2022a) who attributed it to changes in the
atmospheric circulation. Moreover, Parajuli et al. (2022, 2023) utilized high-resolution WRF-
Chem simulations and advanced aerosol schemes to analyse the dust and rainfall dynamics,
providing insights into the direct and indirect effects of dust on rainfall, which aids in better
regional water resource planning through accurate rainfall predictions. In particular, while
through the indirect effects dust promotes precipitation provided there is sufficient moisture
for both normal and extreme rainfall events, the dust direct effects on precipitation shift from
negative for normal rainfall events (weaker sea-breeze arising from surface cooling) to positive
in extreme events (smaller effects on the sea breeze). Zhang et al. (2024) stressed the two-way
interaction between dust aerosols and the Planetary Boundary Layer (PBL) dynamics: aerosols
directly impact the PBL structure through direct and indirect effects, while the the modified
PBL characteristics and low-level circulation modulate aerosol processes. All the
aforementioned studies focus on dust aerosols, there is no assessment to date of the model
performance for the simulation of gaseous pollutants over the region. This is crucial, given the
complex dynamics between anthropogenic and natural factors in air quality management and
the necessity of tailored model configurations for accurate environmental assessments in arid
regions.

This study represents the first comprehensive evaluation of the WRF-Chem model in the
Arabian Peninsula, with a focus on the UAE, a country that is representative of those in the
region, specifically examining concentrations of air pollutants along with crucial
meteorological parameters relevant to air quality studies. The primary objective of this study
is twofold:

● Evaluate the WRF-Chem's ability to replicate meteorological conditions. This involves
comparing the model's simulation of temperature, wind speed, relative humidity,
downward short-wave radiation and boundary layer height against ground-based
observations and data from the European Centre for Medium-Range Weather
Forecasting (ECMWF) fifth reanalysis product, ERA5 (Hersbach et al., 2020);
● Assess the model's performance in simulating concentrations of key gaseous pollutants,
specifically $NO_2$, $O_3$, and CO, which are prevalent in the region (Teixido et al., 2021),
against data from the TROPOspheric Monitoring Instrument (TROPOMI; Veefkind et
al., 2012) onboard the Sentinel-5 Precursor (S5P) satellite. Additionally, aerosol optical
depth (AOD) at 550 nm from AERONET and MODIS satellite observations are used
to evaluate the model's skill in simulating aerosol concentrations.

The structure of the paper is as follows. Section 2 describes the configuration of the WRF-
Chem considered in this work. Section 3 elaborates on the methodology and datasets used in
this study. Section 4 provides a comprehensive assessment of the model's simulated data
against observational datasets, reanalysis and satellite-derived products. Section 5 concludes
by outlining the main findings.

## 2. WRF-Chem configuration

WRF-Chem version 4.3.1 is employed to simulate the atmospheric conditions and transport
of pollutants in the UAE. WRF-Chem is a mesoscale regional chemistry transport model,
developed by the National Oceanic and Atmospheric Administration (NOAA) Earth System
Research Laboratory (ESRL), with contributions from the global science community. In WRF-
Chem, the air quality and meteorological components are predicted simultaneously using the
same grid, transport, timestep, and sub-grid scale physics. A detailed description of the model
is found in Grell et al., (2005), Skamarock et al., (2008) and Powers et al., (2017). The physics
schemes employed in the simulations are the Rapid Radiative Transfer Model for Global
Circulations Models (RRTMG) for radiation parametrization of both short and long wave
radiation (Iacono et al., 2008), the cloud microphysics is represented by the Morrison 2-
moment (Morrison et al., 2009), and the Kain-Fritsch scheme is used for convective
parameterisation (Kain, J.S, 2004) with the subgrid-scale cloud feedback to radiation switched
on (Alapaty et al., 2012). The Unified Noah model is used to represent the land surface model
(Tewari et al., 2004), with an improved representation of soil texture and land use/land cover
(LULC) over the UAE (Temimi et al., 2020). The boundary layer dynamics are represented by
the Yonsei University (YSU) scheme (Hong, 2010). The chosen physics schemes are listed in
Table 1. The simulated mesoscale meteorology is kept in line with the analysed meteorology
through spectral nudging to the National Centre for Environmental Prediction (NCEP) Global
Forecast System (GFS) analyses used to drive the model, in an attempt to limit errors in the
mesoscale transport. During the simulations, horizontal and vertical wind, potential
temperature and water vapour mixing ratio are nudged to GFS analyses in all model layers
above the planetary boundary layer on a time-scale of 6 hours for scales above ~1000 km.
Meteorological conditions were initialised by NCEP GFS 6-hourly analyses at 0.25° resolution.
This study utilised the Model for Ozone and Related Chemical Tracers, version 4
(MOZART-4) chemical mechanism for calculating gas-phase chemistry, which includes 81
chemical species with 159 gas-phase reactions and 38 photolysis processes (Emmons et al.,
2010). Aerosol chemistry is represented by the GOCART (Chin et al., 2002) module, along
with the Tropospheric, Ultraviolet and Visible (TUV) full photolysis scheme (Madronich,
1987; Tie, 2003), which deploys climatological $O_3$ and $O_2$ columns. Dry deposition is
calculated using Wesely (1989). Anthropogenic emissions are taken from the Emission
Database for Global Atmospheric Research (EDGAR) version 8.1 at a $0.1 \times 0.1°$ horizontal
resolution for 2022 (Crippa et al., 2020), consistent with the simulation period. Emissions
include $SO_2$, NOx, CO, Non-Methane Volatile Organic Compounds (NMVOC), $NH_3$, black
carbon (BC) and organic carbon (OC). Biogenic emissions are calculated online by the Model
of Emissions of Gases and Aerosol from Nature (MEGAN; Guenther et al., 2012). The
chemistry boundary conditions (BCs) used in domain D01 and the initial conditions (ICs) for
all domains in the WRF-Chem simulations are extracted from CAM-chem model forecasts
(Emmons et al., 2020). In this work, we run the WRF-Chem model on the three nested domains
with horizontal resolutions of 27-, 9-, and 3-km corresponding to 283×205, 271×193, and
256×178 grid points, respectively. In the vertical, there are 45 layers, with the lowest model
level at about 27 m above the surface. The outermost domain covers most of the Middle East
and the surrounding region, while the innermost domain covers the entire UAE (Fig. 1(a)). The
analysis in this research article exclusively utilizes results from the inner domain (D03). Fig.
1(b) shows the spatial distribution of UAE airport stations, the WISE-UAE observational site,
and AERONET locations for AOD measurements.

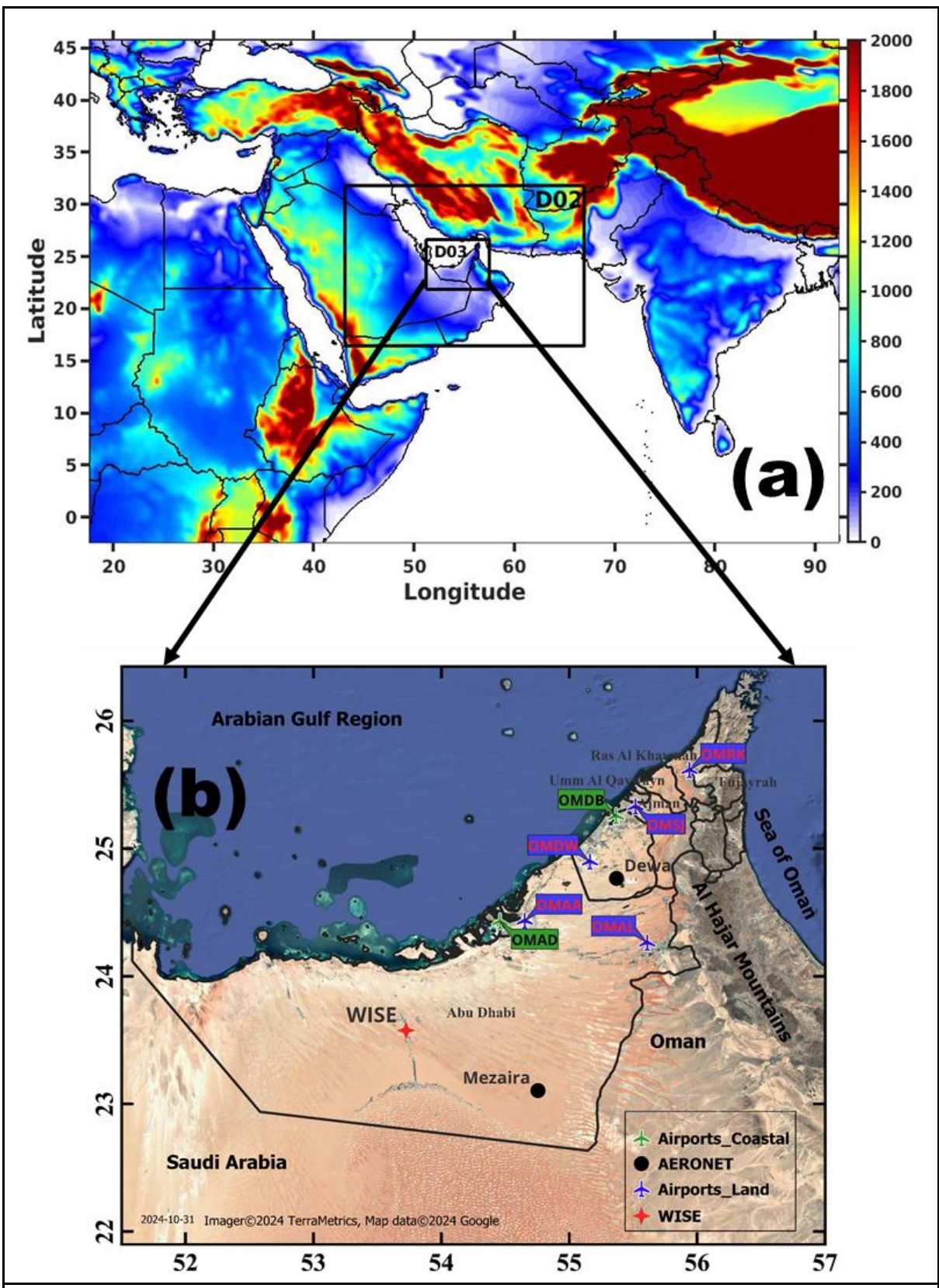

**Figure 1: Model Configuration:** (a) The WRF domain configuration consists of three telescoping nests, with the outermost boundaries denoting the parent grid (D01). D02 and D03 are the nested domains. Bottom panel (b) is a zoom of the innermost domain (D03)

showing the spatial distribution of the seven automatic weather stations operated in airports (land stations (5) are denoted by blue color, coastal stations (2) are represented by green color) along with WInd-blown Sand Experiment (WISE)-United Arab Emirates (UAE) Site by reg color star and black dots represent two AERONET stations (Mezaira and Dewa). The shading in (a) represents the orography (m). Further details about the stations are given in Tables S1.


The WRF-Chem simulation is driven by anthropogenic emissions from the EDGAR database, version 8.1, at a horizontal resolution of $0.1° \times 0.1°$ for the year 2022 (Crippa et al., 2020). The EDGAR emission inventory accounts for day-to-day variability (e.g., weekday versus weekend) and hourly fluctuations (diurnal cycle) of anthropogenic emissions, as detailed by Crippa et al. (2020). For example, road transport emissions are generally lower at night and higher during daytime hours, while agricultural emissions tend to peak during specific months. To achieve an hourly resolution for the model, we scaled the coarsely resolved emission data using predefined hourly, daily, and monthly scaling factors (temporal profiles). The initial temporal profiles are derived from the work of Olivier et al. (2003) and have been refined to place greater emphasis on the most relevant emission sectors for each pollutant within the study region. According to the Environment Agency – Abu Dhabi (2018), the primary sectors contributing to emissions include traffic, the power industry, energy used in buildings, and the manufacturing industry. Using these optimised emission profiles, emissions for $NO_2$ and CO were dynamically adjusted during the model simulations to better capture local emission patterns and their variability. However, the results indicated that emissions for $NO_2$ and CO are underestimated by EDGAR. Although WRF-Chem simulations incorporate temporal profiles of emissions, the impact of these emission estimates on daily variations could not be fully assessed in this study due to the lack of ground-based measurements and the limited temporal resolution of satellite data. MODIS and TROPOMI satellites each pass over the study area only once per day, restricting the ability to capture daily variations comprehensively. Consequently, this article is limited in its assessment of daily emission variability. Moreover, WRF-Chem supports the vertical distribution of trace gas emissions, which is particularly useful for capturing emissions released at elevated altitudes, such as those from combustion stacks. Accurately representing the vertical distribution of emissions is important for simulating atmospheric processes. However, incorporating this complexity would likely provide minimal improvements in model accuracy for regions where surface emissions

dominate, and where observational constraints are largely limited to coarse vertical resolution
or surface-level data. Therefore, in this study, all emissions were injected into the lowest model
layer to align with the observational data characteristics and the typical conditions in the study
area.
**Table 1: WRF-chem model setup**

| Model set-up | Option |
|---|---|
| Model version | 4.3.3 |
| Domain | 3 domains |
| Horizontal resolution | D01:27km, D02:9km and D03:3km |
| Simulation period | Monthly runs from June and December 2022 |
| Model spin-up period | 2 days in each month |
| Vertical resolution | 45 eta levels up to 50 hPa. |
| Domain size | D01: 283×205 grids, D02: 271×193 grids and D03: 256×178 grids |
| Meteorological boundary | NCEP FNL reanalysis (0.25º, 6-hourly) |
| Chemical boundary | CAM-Chem (Emmons, Fasullo, et al., 2020) |
| **Physical Process** | **Parameterization Scheme** |
| Microphysics | Morrison double moment (Morrison et al., 2009) |
| Cumulus parameterization | Kain-Fritsch (Kain, 2004) with the subgrid-scale cloud-radiation feedbacks activated (Alapaty et al., 2012) |
| Shortwave radiation | Rapid Radiative Transfer Model for GCMs (RRTMG) (Iacono et al., 2008) |
| Longwave radiation | Rapid Radiative Transfer Model for GCMs (RRTMG) (Iacono et al., 2008) |
| Land surface | Unified Noah land surface model (Tewari et al., 2004) |
| Planetary boundary layer | Yonsei University scheme (Hong, 2010) |
| **Chemistry option** | **Scheme used** |
| Gas phase chemistry | MOZART-4 (Emmons et al., 2010) |
| Aerosol chemistry | GOCART (Chin et al., 2002) |

| | |
|---|---|
| Photolysis | Madronich F-TUV (Madronich, 1987; Tie, 2003) |
| Biogenic emissions | MEGAN (Guenther et al. 2012) |
| Dry deposition | Wesely (Wesely 1989) |


## 3. Data Sets and methodology

### 3.1 Meteorology observations

In this study, meteorological data from 8 automatic weather stations (AWS) operated at UAE airports are utilized to assess the WRF-Chem air temperature at 2 meters above ground (T2m), wind speed at 10 meters (WS10m), and relative humidity at 2 meters above ground (RH2m) forecasts during June and December of 2022. The spatial distribution of the stations across the UAE is illustrated in Fig. 1(b) (refer to Table S1 for more details). These locations are categorically divided into two regions—land stations (station codeL OMAA, OMDW, OMAL, OMSJ, OMRK) and coastal stations (station code: OMAD, OMDB)—following the criteria outlined in Branch et al., (2021). Subsequent analyses are based on these two primary categories, with the land region comprising 5 stations and the coastal region comprising 2 stations (Fig. 1b). In addition to the UAE airports data, we utilized meteorological data from the WInd-Blown Sand Experiment (WISE)-UAE measurements. The WISE-UAE experiment started on 25 July 2022 at Madinat Zayed (23.5761°N, 53.7242°E; elevation: 119 m; Fig. 1b), located 120 km southwest of Abu Dhabi, UAE. An overview of the instrumentation and experiment site used during WISE-UAE is provided in Nelli et al. (2024(a, b)). This study uses WS10m T2m, RH2m, and downward shortwave radiation flux (SW) from these measurements to validate the WRF-Chem simulations for December 2022. The specifications and accuracies of the instruments used in WISE-UAE are outlined in detail, along with the stringent quality control procedures applied, as described in Nelli et al. (2024(a,b,c)).

### 3.2 AERONET

The Aerosol Robotic Network (AERONET) program is a global federation of ground-based sun photometers comprising more than 400 stations worldwide (Holben et al., 1998). AERONET utilizes multiple bands ranging from UV to near-IR wavelengths to measure spectral sun irradiance and sky radiances, from which Aerosol Optical Depth (AOD) at 550 nm and other aerosol properties are derived. A detailed description of the AERONET retrievals is

provided in Holben et al. (1998). This study uses Level 2.0 AOD data at 550 nm from Mezaira
for June and from Dewa for December 2022, with an hourly resolution. It is important to note
that AOD retrieved from AERONET is accurate to within 0.01 (Dubovik et al., 2000).

## 3.3 ERA-5 Reanalysis data

The fifth-generation ECMWF reanalysis, known as ERA-5 (Hersbach et al., 2020),
represents a significant advancement over its predecessor, ERA-Interim, introduced by Dee et
al. (2011). ERA-5 incorporates a sophisticated four-dimensional variational (4D-Var) data
assimilation method, utilizing the 41r2 cycle of the Integrated Forecast System (IFS). This
system is enhanced by integrating a soil and ocean wave models, offering a comprehensive
approach to climate data analysis. We accessed ERA-5 data through the Copernicus Climate
Change Service Climate Data Store (CDS) for this research. The dataset provides atmospheric
observations across 137 hybrid vertical levels, with raw model data interpolated onto 37
distinct pressure levels. These levels span from 1000 hPa, close to the Earth's surface, up to 1
hPa, reaching altitudes of approximately 80 km. Further details on the ERA-5 dataset are
available in Dee et al. (2011) and Hersbach et al. (2020). Our study utilizes explicitly hourly
data for a selection of meteorological parameters: T2m, WS10m, SW, and planetary boundary
layer height (PBL), for June and December 2022.

## 3.4 Satellite-borne observations: TROPOMI

Launched by the European Space Agency (ESA) on October 13, 2017, the TROPOMI
instrument is aboard the S5P satellite, operating in a near-polar sun-synchronous orbit.
Positioned at an altitude of 817 km, the S5P satellite crosses the equator at a local solar time of
13:30, boasting a wide swath of approximately 2600 km, and providing daily global coverage.
TROPOMI features four distinct spectrometers that measure the radiation in the ultraviolet
(UV) and UV-visible (UV-VIS) range (270 to 500 nm), near-infrared (NIR) range (675 to 775
nm), and short-wave infrared (SWIR) range (2305 to 2385 nm) spectral bands (Veefkind et al.,
2012). Notably, the last two spectral bands, NIR and SWIR, are newly introduced in
TROPOMI compared to its predecessor OMI (Ozone Monitoring Instrument). TROPOMI's
data products encompass daily observations of trace gases, including CO, $O_3$, $NO_2$, $CH_4$,
HCHO, aerosols, and cloud properties. This study utilized daily tropospheric $NO_2$, total CO
columns, and ozone profile level 2 products downloaded from the GES DISC website
(https://disc.gsfc.nasa.gov/) for the period of June and December 2022. The specific data sets
employed for the present study include S5P_OFFL_L2__O3__PR for $O_3$,
S5P_OFFL_L2__CO for CO, and S5P_OFFL_L2__NO2 for $NO_2$, covering the study region
bounded by longitudes [51°,58°] and latitudes [21°,27°]. Further details on each product,
including the retrieval algorithms and validation results, are summarized in the following
section.

TROPOMI retrieval of $NO_2$ columns are derived using UV-VIS spectrometer backscattered
solar radiation measurements in the wavelength range of 405-465 nm and provides total and
tropospheric $NO_2$ vertical column density with a near-nadir resolution of $7 \times 3.5$ km. The total
$NO_2$ slant column density (SCD) is retrieved from the measured solar irradiance spectra using
the Differential Optical Absorption Spectroscopy (DOAS) method. Tropospheric and
stratospheric slant column densities are separated from SCD by a data assimilation system
based on the chemistry transport model V5 (TM5-MP). Afterwards, they are converted to
vertical column densities (VCDs) with the help of look-up table of altitude-dependent air-mass
factors (AMFs) and information on the vertical distribution of $NO_2$ from TM5-MP apriori
profile with a horizontal resolution of 1° x 1° and a time step of 30 min (Boersma et al., 2018;
Van Geffen et al., 2022). The TROPOMI $NO_2$ product has been extensively evaluated using
ground-based and aircraft observations and is found to have a high correlation and low bias of
less than 30% with respect to in-situ measurements (Griffin et al., 2019; Ialongo et al., 2020).
We used the both reprocessed (RPRO) and offline (OFFL) TROPOMI $NO_2$ data files from the
most recent processor versions depending on availability for a given day of observations.
Additionally, there is another $NO_2$ product available as near-real time (NRTI). NRTI data files
are generated using TM5-MP forecast data rather than analysis data as with REPO and OFFL
files (Van Geffen et al., 2022). The differences between the OFFL/REPO and NRTI $NO_2$
products are generally very small (Ialongo et al. (2020) and references therein).

The Shortwave Infrared Carbon Monoxide Retrieval (SICOR) algorithm is used to retrieve
CO total column densities from TROPOMI in the spectral range of 2305 to 2385 nm (Landgraf
et al., 2016). The SICOR algorithm accounts for a profile-scaling approach that scales retrieved
CO total column to the a priori reference profile. The a priori reference profiles are taken from
the global chemistry transport model simulations of TM5-MP, and vary based on the location,
month and year (Krol et al., 2005). The detailed outline of all settings and other auxiliary data
sets used for CO retrievals is given in Landgraf et al., (2016). This study limits the analysis to
CO pixels corresponding to clear-sky conditions and mid-level clouds by filtering the data

using the quality flag variable (qa_value). The scenes corresponding to qa_value > 0.5 are used in this current analysis as suggested in the ATBD (algorithm theoretical baseline document; Landgraf et al., 2016). In this present work, TROPOMI CO measurements for June and December 2022 have been analysed. Moreover, we use either the reprocessed (RPRO) or offline (OFFL) data files from most recent processor versions depending on availability for a given day of observations. Wizenberg et al., (2021) compared global TROPOMI retrieved CO total columns with corresponding ACE-FTS (Atmospheric Chemistry Experiment- Fourier transform spectrometer) columns for the period from November 2017 to May 2020 and found a small relative bias of -0.83% with a correlation coefficient of 0.93 between two data sets. Similar results are also found between TROPOMI CO with corresponding CO fields from the ECMWF assimilation system: Borsdorff et al. (2018) reported a small mean difference between the two data sets of 3.2% with a correlation coefficient of 0.97.

TROPOMI also provides ozone profiles (5P_OFFL_L2__O3__PR) at 33 pressure levels with a horizontal resolution of 28x28 km. It measures radiances and irradiances in the ultraviolet wavelength of 270-330 nm and provides the ozone profile information. The Optimal Estimation (OE) algorithm is used to retrieve the ozone profile data. Before this stage, various pre-processing steps are applied to the measured spectra before the estimation of the ozone profile. The main process of the algorithm is the OE method, which combines the information from the measured spectra with the a-priori information. The latter is based on climatology as described in Labow et al., (2015). The description of the various pre-processing steps performed to retrieve ozone profiles is presented in the Algorithm Theoretical Basis Document (Veefkind, et al., 2021). The validation of TROPOMI retrieved ozone profile data against the ground-based measurements reported a median bias of 0.3% for OFFL/REPO products while 0.8% for NRTI ozone products (Lambert et al., 2023). Our focus is specifically on the tropospheric ozone column due to its direct relevance to surface air quality. Total column ozone measurements are primarily influenced by stratospheric ozone, which accounts for approximately 90% of the total column, while tropospheric ozone comprises only around 10%. Given this, we have used ozone profile data from the surface to 100 hPa., designated as tropospheric ozone columns for this study and referred to as TROPOMI-$O_3$, expressed in Dobson Units (DU), where 1 DU = $2.69 \times 10^{16}$ molecules/cm².

## 3.5 Satellite-borne observations: MODIS

The Moderate Resolution Imaging Spectroradiometer (MODIS) sensor was launched into the polar sun-synchronous orbit at an altitude of 705 km aboard NASA's two Earth Observing System (EOS) satellites, Terra (Feb-2000) and Aqua (June-2002) [Kaufaman et al., 1997; Remer et al., 2005] . The equator crossing times of two satellites were, Terra crossing at 1030 LST and Aqua crossing at 1330 LST. The MODIS sensor has a swath of ~2330 km and provides near-global coverage with a temporal resolution of 1-2 days. The sensor measures the reflected solar radiation from the Earth's atmosphere and the surface as well as emitted thermal radiation at 36 spectral bands from 0.41 to 14 µm with three spatial resolutions: 250m, 500m, and 1km. Seven of these bands operating in the spectral range of 0.415-2.155 µm can effectively retrieve the AOD over land and ocean [Levy et al. 2013 ; Hsu et al. 2015 ; Sayer et al., 2014a; 2014b; 2015] . The MODIS retrieval algorithm is based on the lookup table approach with a pre-defined set of aerosol types, loadings and geometries [Floutsi et al. 2016]. A comprehensive description of retrieval algorithms and details of MODIS instrument are found elsewhere [Remer et al. 2008; Levy et al. 2013 ]. MODIS AOD retrieval algorithms have been substantially validated against in-situ and/or other remote sensing data sets from regional to global scales and are updated periodically [Remer et al. 2008 ; Li et al. 2009]. The uncertainty of AOD retrievals is estimated to be ±0.05±0.20 x AOD over land and ±0.03±0.15 x AOD over ocean [Remer et al.,2005; 2008]. The present study utilized Level 2 MODIS aerosol products (Collection 6.1) obtained from the Atmosphere Archive and Distribution System (LAADS DAAC). These products consist of 5-minute satellite swaths with a spatial resolution of 10 km, covering the period of June and December 2022. (Devadiga, 2024).

## 3.6 Satellite data processing

In order to quantitatively compare the WRF-chem simulations with satellite measurements, the model outputs must be processed using the appropriate method as described in the literature (Kumar et al., 2012). Direct comparison between satellite retrievals and model outputs is not recommended, as satellite measurements depend on column averaging kernels (AK) and a-priori profiles. The AK vector represents the vertical sensitivity of the retrieved column relative to the true vertical profile of the target variable in the atmosphere. It indicates how changes in the true atmospheric profile at different vertical levels influence the retrieved column values, allowing for a more accurate comparison between model simulations and TROPOMI data by

convolving the model outputs with the AK. The typical AK vectors are plotted over the WISE-UAE location to know the sensitivity of AK at different pressure levels (Figure S7)

The column density from the WRF-Chem model is re-gridded to match the TROPOMI instrument's grids and is vertically interpolated to the TROPOMI pressure levels before it is multiplied by the AK. This treatment of the WRF-Chem-simulated profile with the column averaging kernels allows for a comparison that is independent of the chemical transport model (CTM) a-priori assumptions and the vertical sensitivity of the retrieval process; therefore, it can be directly compared with the TROPOMI-derived tropospheric column of $NO_2$. The TROPOMI-$NO_2$ and TROPOMI-CO products also provide a column averaging kernel matrix. In this case the column AK averaging kernel accounts for the vertical distribution and sensitivity of the measurements, as classically done by Borsdorff et al., (2014) as:

$$X_{ret} = X_{a\ prior} + AK \times (X_{true} - X_{a\ prior}) + e_x \text{-----------------------------------------------}(1)$$

where $X_{true}$ is model simulation profile of trace gas; $X_{ret}$ is the retrieved profile or smoothed model profile; $e_x$ represents the error on the retrieved trace gas profile; $X_{a\ prior}$ is the a-priori information provided in the TROPOMI data set. For TROPOMI-$NO_2$ data, the contribution of the a priori profile and error on the retrieved profile can be eliminated, as explained in Borsdorff et al., (2014). In particular, eq. (1) simplifies to

$$X_{ret} = AK \times (X_{true}) \text{------------------------------------------------------------------------}(2)$$

where $X_{true}$ represents WRF-Chem simulation profile for both $NO_2$ and CO, AK represents the averaging kernels information provided in the TROPOMI data set for $NO_2$ and CO and $X_{ret}$ represents smoothed model profile for $NO_2$ and CO.

For validation of ozone, we have used the TROPOMI ozone profile level 2 data product S5P_OFFL_L2__O$_3$__ that provides the ozone concentrations at 33 pressure levels. This data product also includes the a priori information and column averaging kernel for each pressure level. In order to compare our model profile with the one given by this dataset, the model output is horizontally and vertically interpolated to TROPOMI grids and vertical levels. The final model profile was calculated by the Eq. (3)

$$X_{ret} = X_{a\ prior} + AK \times (X_{true} - X_{a\ prior}) \text{-----------------------------------------------}(3)$$

where $X_{true}$ represents WRF-Chem simulation profile for O$_3$, AK represents the averaging kernels information provided in the TROPOMI data, $X_{ret}$ represents smoothed model profile for O$_3$ and $X_{a\ prior}$ is the a-priori information provided in the TROPOMI data. Since the highest vertical level in WRF-Chem-simulated trace gas concentration is 50 hPa, the remaining vertical

layers of ozone and CO are made equal to the a priori concentration of respective trace gases as described by ATBD (Landgraf et al., 2016).

## 3.7 Evaluation methodology

Meteorological parameters from the WRF-Chem model are extracted for the grid points closest to the surface observation sites of the AWS. As noted before, the meteorological parameters are categorized and averaged for land and marine regions separately for the regional analysis. To enable the comparison of atmospheric column data from the TROPOMI satellite retrievals with WRF-Chem outputs, the data must undergo smoothing through an appropriate method described in Section 3.4, as direct comparison between satellite retrievals and simulations is not feasible due to discrepancies highlighted in previous literature. Additionally, and owing to the spatial resolution differences between WRF-Chem and ERA5 datasets, it is necessary to remap the model data to the ERA5 grids for accurate comparison. A wide range of statistical parameters is available for evaluating model simulations. In this study, we employed statistical skill scores including the Pearson correlation coefficient (r), the Mean Bias (MB), the Root Mean Square Error (RMSE), and the Mean Absolute Error (MAE), which have been extensively discussed and applied in similar contexts (Fonseca et al., 2021; Ivatt & Evans, 2020; Temimi et al., 2020b).

The following equations (eq. 4 to eq. 7) are used to calculate these statistical matrixes in the present study,

$$r = \frac{\sum_{i=1}^{N}[(O_i - \underline{O_i})(M_i - \underline{M_i})]}{\sum_{i=1}^{N}(O_i - \underline{O_i})^2 \sum_{i=1}^{N}(M_i - \underline{M_i})^2} \text{----------------------------------------------(4)}$$

$$\text{RMSE} = \left(\frac{1}{N}\sum_{i=1}^{N}(M_i - O_i)^2\right)^{\frac{1}{2}} \text{------------------------------------------(5)}$$

$$\text{MB} = \frac{1}{N}\sum_{i=1}^{N}(M_i - O_i) \text{------------------------------------------------(6)}$$

$$\text{MAE} = \frac{1}{N}\sum_{i=1}^{N}|M_i - O_i| \text{----------------------------------------------(7)}$$

where $O_i$ denotes the i-th observation, $M_i$ represents the corresponding WRF-chem model simulated value, and $N$ is the number of model and observation pairs. $\underline{M_i}$ and $\underline{O_i}$ are the model and observational means (i.e. average of 1-30 June and 1-31 December), respectively. The correlation coefficient (r) is an indication of the phase agreement between the modelled and

observed time-series. The RMSE measures the average error in the model predictions, while
the MAE determines the mean error between the model forecasts and observations regardless
of whether it is an under or overestimate. The MB is a measure of the systematic error and
gives information as to whether the model is over or underpredicting the corresponding
observed values.

## 4. Results and Discussion

### 4.1 Model performance for key meteorological variables

The capability of the WRF-Chem model to reproduce realistic spatiotemporal patterns of
key meteorological variables has been assessed by comparing the model outputs to
observational reanalysis data for June and December 2022, representing contrasting summer
and winter conditions over the UAE. Evaluating the accuracy of WRF-Chem's meteorological
forecasts in the study area is essential before applying the model forecasts to air quality
assessments. Accordingly, we compared the model predictions for T2m, RH2m, WS10m, and
SW against ground-based observations at seven airport stations and *in-situ* measurements from
the WISE-UAE field campaign (details in Table S1). Additionally, the boundary layer height
is evaluated against ERA5 reanalysis data, which offers a spatial resolution of approximately
28 km, higher than the other currently available reanalysis datasets. Detailed results of this
analysis are presented in the supplementary material, with key findings summarized here to
support the paper's discussion. The aforementioned meteorological parameters are selected,
given their critical role in influencing air pollutant behavior (Ritter et al., 2013).

### 4.1.1 Evaluation against *In-Situ* Observations

The WRF-Chem model evaluation against observations across the seven meteorological
stations (Table S1) at the UAE airports for T2m, RH2m, and WS10m during June and
December 2022 reveals a close agreement between the modeled and observed values (Table
S2). The cold bias reported by several studies, including Branch et al. (2021), Temimi et al.
(2020a), and Abida et al. (2022), which occurs primarily at night, is reduced in the WRF-Chem
simulations presented here. In fact, and for the June month, the air temperature bias is positive,
~0.2 °C. This stresses the importance of properly simulating the observed aerosol loading in
this hyper arid region. Deficiencies in the land surface model and radiation schemes and in the
representation of the surface properties, particularly the surface emissivity that may be
overestimated in the model (Parajuli et al., 2023), can also account for this discrepancy. The
WRF-Chem model also exhibits a noteworthy dry bias in this region, linked to an incorrect
simulation of the soil moisture and the mesoscale land-sea breeze circulation, which is present
in both seasons. The strength of the near-surface wind speed tends to be overestimated in WRF-
Chem in the UAE by about 1-3 m/s, which has been attributed to an incorrect representation of
its subgrid-scale variability and deficiencies in the surface drag parameterization scheme (Nelli
et al., 2020; Fonseca et al., 2020; Temimi et al., 2020b). Here, the biases are much smaller,
within 0.5 m/s. This, together with the improved representation of the observed air temperature,
reflects an overall improved simulation of the boundary layer dynamics in the model.

The WRF-Chem model evaluation against WISE-UAE measurements (detailed in Table
S3 and Fig. S1) reveals a comparable performance to that seen concerning the seven airport
stations. SW observations are also available for this site. An evaluation against the WRF-Chem
values reveals the model overestimates the incoming shortwave radiation flux by about 30
$W/m^2$ for December, which can be attributed to reduced cloud cover, a known WRF deficiency
(Wehbe et al., 2019; Fonseca et al. 2020, 2022a). An inspection of the diurnal cycle revealed
the cold (typically by 2-3 °C) and dry (by about 20%) biases occur mostly at night, when the
wind speed in the model is higher than that observed, suggesting increased advection of cooler
and drier desert air into the site.

### 4.1.2 Evaluation against ERA5 reanalysis data

The WRF-Chem model predictions are also evaluated against ERA5 reanalysis data for
T2m, WS10m, SW, and PBL during June and December 2022. The air temperature biases are
within 1 °C, with a cold bias present in both months, more pronounced over inland areas, with
correlation coefficients 0.9 (Fig. S2). It is important to note that ERA5 overestimates the
temperature at night and underestimates it during the day typically by 1-2 °C in the country for
all seasons (Nelli et al., 2024a), meaning the cold bias shown by WRF-Chem does not
necessarily indicate a poorer performance. The skill scores for WS10m and SW are also similar
to those estimated concerning the station observations and the WISE-UAE field measurements.
For the PBL height, the model reproduces its spatial and seasonal variations (Fig. S3), largely
driven by the temperature seasonal cycle (cf. Figs. S2; Basha et al., 2019). The PBL height,
and over land areas, ranges from 2400-2500 m in the summer during the day to less than 500
m in winter at night. Over the Arabian Gulf, the PBL is deeper in the winter months in both
ERA-5 and WRF-Chem (800 m vs. 200 m), owing to stronger winds and enhanced turbulent
mixing (Dai, 2024).
This comprehensive evaluation of the predicted meteorological parameters against those
observed at seven UAE airport sites, the WISE-UAE experimental site, and ERA5 reanalysis
data demonstrates that WRF-Chem reliably captures them, including their spatial and seasonal
variations across the UAE. As WRF-Chem integrates meteorological and chemical processes,
precise meteorological simulations are essential to ensure accurate chemical computations
within the model domain.

## 534 4.2 Model performance for the gaseous pollutants

The study incorporates comparative assessments with satellite data from the TROPOMI
instrument, including evaluations of the tropospheric column of $NO_2$ (denoted as TROPOMI-
$NO_2$), total column CO (TROPOMI-CO), and tropospheric column ozone (TROPOMI-$O_3$) for
the corresponding periods within the UAE. The satellite overpass takes place daily at 13:30
local time; therefore, model simulations corresponding to this time are utilized here for
comparison over the study area. After smoothing the model concentrations using the a priori
and averaging kernel matrix, as detailed in Section 3.4, the results are compared with the
corresponding TROPOMI products.

In the troposphere, nitrogen oxides ($NOx = NO + NO_2$) are vital for ozone production and
depletion processes in sunlight. Due to their relatively short lifespan, NOx concentrations are
closely linked to emission sources, making them highly sensitive to inaccuracies in emission
estimates compared to other pollutants. In our model setup, we adopt the recommendation of
Emmons et al. (2010), assigning 10% of NOx emissions as $NO_2$. As a result, the model tends
to underestimate TROPOMI $NO_2$ levels, particularly in regions with high emission sources,
such as urban centres. The Environment Agency – Abu Dhabi (2018) reported that oil and gas,
road transport, and electricity generation are the primary sectors contributing to NOx total
emissions, accounting for 42%, 34%, and 13%, respectively, for the base year of 2015 in the
Emirate of Abu Dhabi. Fig. 2 presents the average spatial distributions of absolute differences
between the model-simulated and the TROPOMI-retrieved tropospheric column NO2, scatter
plots, and histograms of relative frequency. The satellite retrievals indicate elevated levels of
$NO_2$ columns, exceeding 5x1015 molecules/cm2, in densely populated industrial areas adjacent
to the major cities of Dubai and Abu Dhabi in both summer and winter (Fig. S4). Conversely,
lower NO$_2$ values, less than 1.5x10$^{15}$ molecules/cm$^2$, are observed over the less urbanized
areas. The higher columns are associated with significant economic development driven by a
high demand in power generation and water desalination projects, which primarily depends on
the combustion of fossil fuels (Abuelgasim & Farahat, 2020; Li et al., 2010). The model
effectively reproduces the spatial distributions of NO$_2$ during the summer and winter of 2022
as depicted in Fig. 2. Even though the biases are positive in rural areas, the observed column
NO$_2$ concentration is underestimated by up to 2x10$^{15}$ molecules/cm$^2$ in the heavily populated
north-eastern UAE, in particular around Ras Al Khaimah and Dubai (Fig. 2a) the sixth-largest
city by population in the country and home to a global ceramic manufacturing company. This
discrepancy suggests that anthropogenic and industrial emissions might be improperly
represented in the EDGAR emission inventory, at least for the UAE. Challenges range from
the incomplete characterization of emissions in source regions to the impact of model
resolution on capturing sub-grid emission sources. Besides deficiencies in the emission
sources, other reasons may explain the model's underperformance in this region. Hoshyaripour
et al. (2016) found that the PBL is shallower and more stable at night when simulated with the
YSU boundary layer scheme used in the WRF-Chem runs, resulting in a higher accumulation
of NOx in the surface layers. As the evaluation conducted here against satellite observations is
daily, an incorrect representation of the atmospheric dynamics will be reflected in the WRF-
Chem predictions. Additionally, the existing model configuration does not include the
formation of secondary aerosols, indicating a potential area for improvement in future versions.
The absence of a vertical distribution of anthropogenic emissions in the model simulations also
plays a pivotal role in these model discrepancies. The satellite retrieved TROPOMI-NO$_2$
averaged for the d03 is 1.1 x 10$^{15}$ molecules/cm$^2$ in summer and 1.03 x 10$^{15}$ molecules/cm$^2$ in
winter, with the corresponding model simulated column concentration of 1.6 x 10$^{15}$ and 1.2 x
10$^{15}$ molecules/cm$^2$, respectively. The model demonstrated a moderate correlation with
satellite-derived NO$_2$ column measurements, achieving correlation coefficients of 0.59 for
summer and 0.58 for winter (refer to Table 2). It tended to overestimate NO$_2$ levels more in
summer, with a discrepancy of 0.5 x 10$^{15}$, compared to 0.2 x 10$^{15}$ molecules/cm$^2$ in winter.
Moreover, the evaluation shows RMSE values of 0.2 x 10$^{15}$ to 0.1 x 10$^{15}$ molecules/cm$^2$ and
MAE values of 0.7 x 10$^{15}$ to 0.5 x 10$^{15}$ molecules/cm$^2$ during the seasons. The frequency
distributions in Fig 2(c) and (f) illustrate the differences in NO$_2$ concentrations between the
WRF-Chem model and TROPOMI observations during summer and winter, respectively. In
panel 2(c), the distribution of differences is entirely positive, indicating that the WRF-Chem
model consistently overestimates NO$_2$ concentrations compared to TROPOMI observations for

592 the summer of 2022. In contrast, Fig. 2(f) shows both positive and negative differences,

593 indicating that the WRF-Chem model exhibits a mix of overestimations and underestimations

594 of NO₂ concentrations in winter, although the majority of differences are still positive. This

595 suggests a more variable alignment between WRF-Chem and TROPOMI-NO₂ in winter, with

596 a general tendency toward overestimation but occasional instances of underestimation.


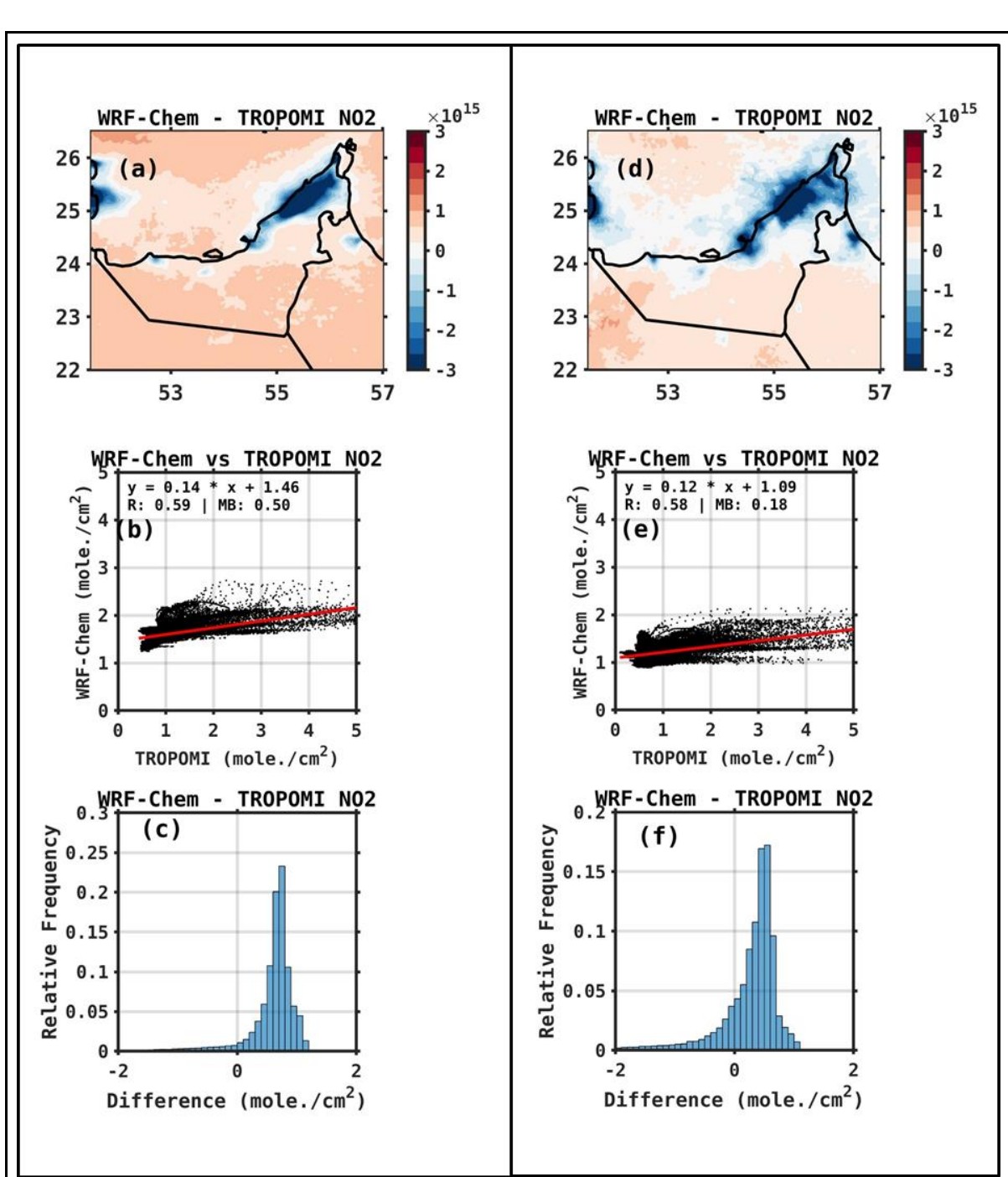

**Figure 2: Evaluation of WRF-chem against satellite-derived NO$_2$**: The average difference between tropospheric column NO2 (mole/cm²) from the TROPOMI satellite and simulated by WRF-Chem, for (a) June and (d) December 2022. (b)-(e) and (c)-(f) are as (a) and (d) but showing scatter plots and histograms of the differences, respectively.


In Fig. 3, the assessment of the model-simulated total column CO and the corresponding
TROPOMI-retrieved values is presented. The statistical metrics comparing these datasets are
provided in Table 2. Fig. S5, shows the comparison of total column CO concentrations over
the domain as observed by the TROPOMI satellite and simulated by the WRF-Chem model (.
Panels (a) and (c) display TROPOMI-CO for summer and winter, showing spatial variations
in CO concentration across the region. High concentrations, particularly over the northern
areas, while lower concentrations found the southern areas. Panels (b) and (d) illustrate
corresponding WRF-Chem CO simulations for the same periods, providing a model-based
estimate of CO distribution. The WRF-Chem model appears to capture the general spatial
patterns observed by TROPOMI, though there may be some discrepancies in the intensity and
precise locations of high CO concentrations. This comparison highlights areas where the WRF-
Chem model aligns well with satellite observations and regions where further adjustments in
model parameters may be necessary to better replicate observed patterns. The TROPOMI-
retrieved CO columns display values of $1.92 \times 10^{18}$ and $1.79 \times 10^{18}$ molecules/cm$^2$ for summer
and winter, respectively. In contrast, the simulated column values are of $1.93 \times 10^{18}$ for summer
and $1.91 \times 10^{18}$ molecules/cm$^2$ for winter. Thus, comparing WRF-Chem and TROPOMI-CO
data reveals more pronounced discrepancies, with a minor overestimation of $0.02 \times 10^{18}$
molecules/cm$^2$ in summer and a significant underestimation of $0.12 \times 10^{18}$ molecules/cm$^2$ in
winter. Shami et al. (2022) found that the EDGAR emissions inventory underestimates CO
emissions when compared to Lebanon's national emission inventory, identifying the road
transport sector as the primary source of CO emissions. Consequently, EDGAR's estimates for
CO emissions are lower than those provided by Waked et al. (2012) for the same region. The
Environment Agency – Abu Dhabi (2018) reported that the road transport sector is the primary
source of CO emissions in Abu Dhabi, accounting for 74% of the total CO emissions.
Additionally, the industrial sector contributes 21% to the total CO emissions. Kumar et al.
(2022) observed an underestimation of CO by the WRF-Chem model, attributing it to an
inaccurate representation of anthropogenic emissions on the vertical scale, not represented in
the current WRF-Chem simulations as noted for NO$_2$. This could result in a more rapid
deposition of CO molecules at the surface, thereby leading to the observed underestimation. In
the summer months, the underprediction of the column CO over coastal areas, in particular
around the major urban centers, and the overprediction over inland regions suggests
deficiencies in the representation of the atmospheric flow (e.g., a too strong onshore flow),
coupled with the aforementioned biases in the emission inventory. In contrast, in winter the
biases are positive, and probably more strongly linked to chemistry than to meteorological
dynamics.
The model output correlates reasonably well with TROPOMI-CO with a correlation coefficient
of 0.82 and 0.40 and an RMSE of $0.03 \times 10^{18}$ and $0.04 \times 10^{18}$ molecules/cm$^2$ in summer and
winter, respectively (Table 2). The frequency distribution in Fig. (c) shows most differences,
with a slight positive skew, suggesting a tendency for the WRF-Chem model to slightly
overestimate CO concentrations compared to TROPOMI observations for summer. In contrast,
Fig. (f) displays a broader distribution with a more pronounced positive skew, indicating larger
and more variable overestimations by WRF-Chem in winter. In winter seasons, the lower
correlation coefficients and higher biases for TROPOMI-CO as compared to TROPOMI-NO$_2$
might be attributed to the complexities inherent in modeling and observing CO distributions,
which local emission sources, atmospheric chemistry, and transport processes can influence.
These findings are consistent with research conducted in India, where Dekker et al. (2019)
reported a correlation of 0.81 between TROPOMI and WRF-Chem CO levels during a high
pollution episode in November 2017. Similarly, in East Asia, Zhang et al. (2016a) documented
correlations between WRF-Chem simulated and the Measurements of Pollution in the
Troposphere (MOPITT)-retrieved CO columns, with a r-value of 0.59 and RMSE of $4.6 \times 10^{17}$
molecules/cm$^2$ for summer, and 0.69 with RMSE of $5.2 \times 10^{17}$ molecules/cm$^2$ for winter,
respectively.

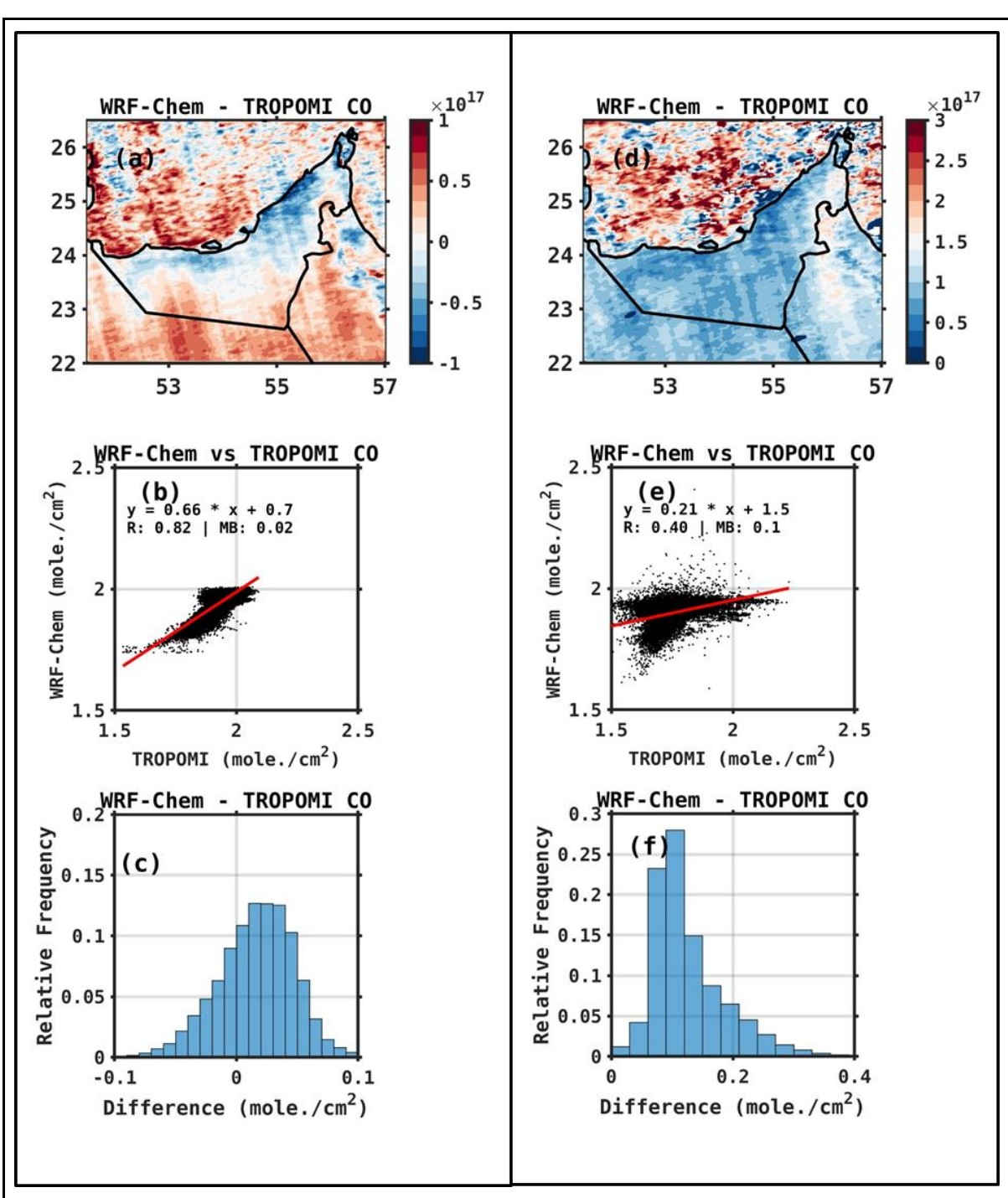

**Figure 3: Evaluation of WRF-Chem against satellite-derived CO:** Same as Fig. 2 but for the total column of CO.


Figure S6 presents the spatial distribution of tropospheric ozone concentrations over the UAE as observed by TROPOMI (TROPOMI-$O_3$) and simulated by the WRF-Chem model during the summer and winter of 2022. In Figures S5 (a) and (b), TROPOMI shows varying $O_3$ concentrations with higher values, particularly along the northern coastal regions, where

concentrations reach up to 20 DU. Similarly, WRF-Chem demonstrates a comparable spatial pattern, with elevated $O_3$ concentrations in the same regions, reaching up to 40 DU, indicating that the model captures the general distribution observed by TROPOMI. In Figure S5 (c), representing winter, TROPOMI exhibits a different distribution pattern, with overall lower $O_3$ concentrations compared to summer. The WRF-Chem simulation in winter also shows a broader distribution of $O_3$, with concentrations reaching up to 25 DU. While the WRF-Chem model aligns reasonably well with TROPOMI observations, discrepancies in concentration levels highlight both the model's ability to replicate seasonal variations and areas where improvements may be needed, especially in the winter months. The comparison of the WRF-Chem simulated tropospheric ozone columns with the TROPOMI-retrieved columns (TROPOMI-$O_3$) is illustrated in Fig. 4, with the statistical comparisons detailed in Table 2. The TROPOMI-$O_3$ columns show higher in summer, at 16.6 DU, and lower values in winter, at 13.4 DU, which is attributed to increased photochemical activity during the summer months (Reddy et al., 2012; Coates et al., 2016; Badia & Jorba, 2015; Abdallah et al., 2018; Baldasano et al., 2011) The WRF-Chem simulations show these variations, with values of 32.8 DU for summer and 24.8 DU for winter, respectively. Therefore, model output is strongly correlated to the TROPOMI-$O_3$ column concentration, with a correlation coefficient of 0.78 and 0.83 and an RMSE (MAE) of 1.4 and 1.0 DU (15.9 and 11.2 DU) during summer and winter, respectively. The WRF-Chem model systematically overestimates ozone levels by 15.9 and 11.2 DU in summer and winter, respectively. The frequency distribution in Fig. 4(c) represents the differences between WRF-Chem and TROPOMI-$O_3$ concentrations during the summer, showing that they are more pronounced with a positive skew. This indicates a consistent tendency for the WRF-Chem model to overestimate $O_3$ concentrations compared to TROPOMI observations in summer. Similarly, Fig. 4(f) displays a frequency distribution for winter with a positive skew and narrower spread, highlighting that WRF-Chem also tends to overestimate $O_3$ concentrations compared to TROPOMI during this season, although with less variability in the overestimations. Therefore, the WRF-Chem model systematically overestimates $O_3$ concentrations throughout the year, with a slightly more consistent bias observed in winter. Hu et al. (2021) highlighted the substantial influence of meteorological factors on ozone production, noting that temperature, relative humidity, and sunshine duration play significant roles in descending order of importance. Strong solar radiation and elevated temperatures enhance photochemical reactions, increasing ozone formation. In comparison with ERA-5 data (Fig. S1) and station data (Table S2), the colder temperatures observed in WRF-Chem, particularly in winter months when tropospheric column $O_3$ biases are less positive (Table 2),

may explain the overestimation of $O_3$ concentrations in the model. Zhang et al. (2020) found
that low wind speeds and high atmospheric pressure can hinder pollutant dispersion, leading to
ozone accumulation, while Lu et al. (2019) observed that high humidity conditions can deplete
$O_3$ through interactions with water vapor and the production of OH radicals. WRF-Chem's
negative RH2m bias against in situ measurements in both summer and winter (Tables S2 and
S3), combined with temperature biases, may contribute to the model's overprediction of $O_3$.
Further exploration of these chemical interactions would require additional sensitivity analyses
beyond this study's scope. Future work should focus on refining model fidelity by improving
the representation of chemical processes and emissions to enhance air quality projections and
deepen our understanding of regional pollution patterns.
The disparities between WRF-Chem and TROPOMI data highlight the intrinsic challenges
in air quality monitoring and prediction. WRF-Chem's limitations may stem from its
dependency on emissions inventories, which, as noted above, can have significant
discrepancies compared to actual emissions, uncertainty in the meteorological forcing data, and
the representation of atmospheric chemistry. TROPOMI, while offering high-resolution
satellite observations, is subject to constraints related to retrieval algorithms and the influence
of atmospheric conditions on measurement accuracy. Liu et al. (2022) identified that
uncertainties in column observations arise from the challenges in differentiating between
stratospheric and tropospheric contributions and uncertainties in the tropospheric air mass
factor and its spectral fitting. Integrating model predictions with satellite observations,
alongside ground-based measurements, is crucial for enhancing our understanding of air
quality dynamics and improving predictive capabilities. This synergistic approach can help
mitigate biases, enhance accuracy, and provide a more comprehensive view of atmospheric
pollutants' distribution over this region.

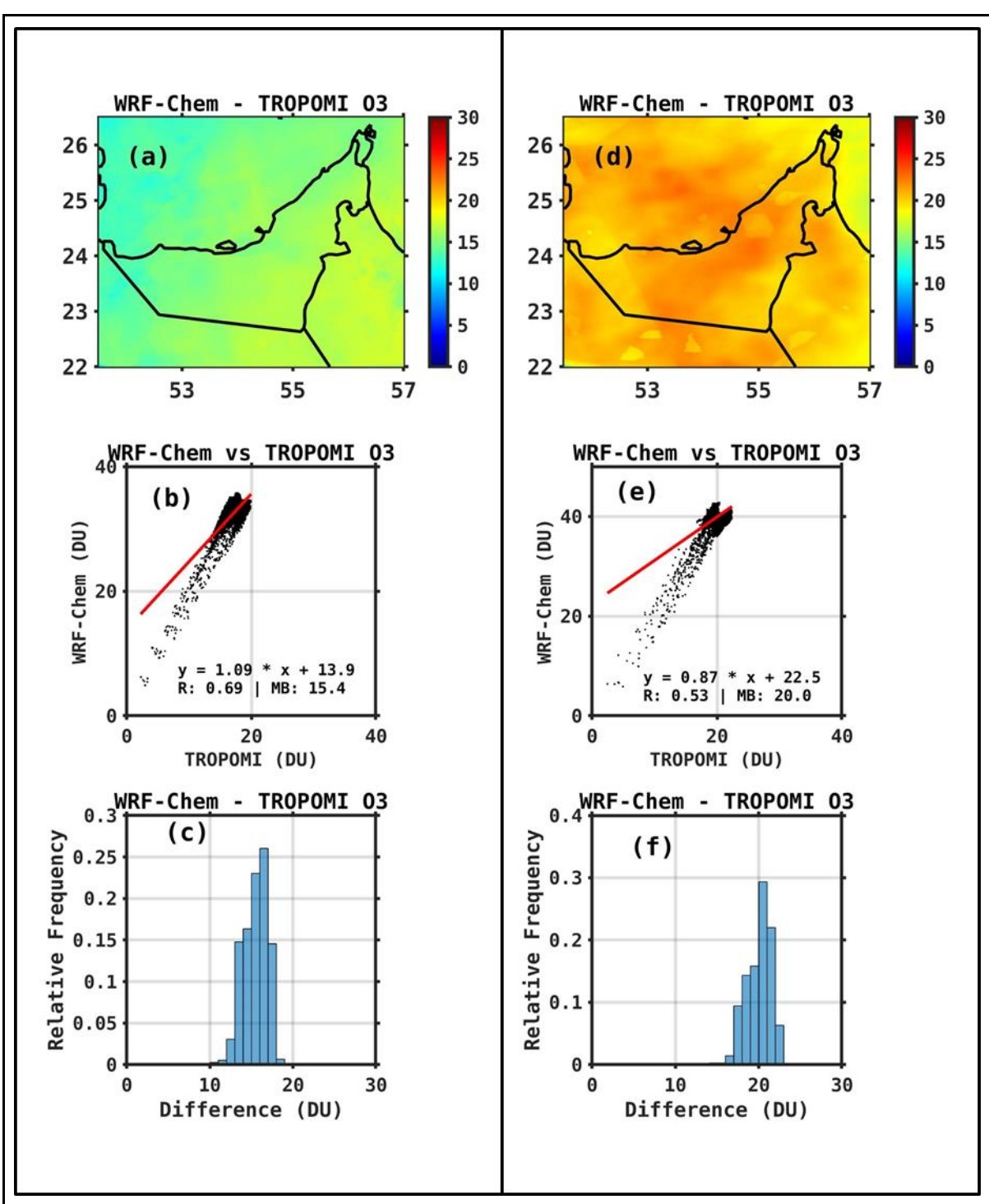

**Figure 4: Evaluation of WRF-Chem against satellite-derived O₃:** Same as Fig. 5 but for total column of ozone.


**Table 2: Statistical verification scores for evaluation against TROPOMI measurements:** skill
scores between TROPOMI columns (mole. /cm$^2$), tropospheric column NO$_2$ (TROPOMI-NO$_2$), total
column carbon monoxide (TROPOMI-CO), tropospheric column ozone (TROPOMI-O$_3$) and MODIS
AOD with corresponding WRF-chem simulated columns during June and December of 2022 over UAE.

The first two columns show the model and satellite monthly-mean values, with the other four giving the MB, MAE and RMSE in units of mole. / $cm^2$ for TROPOMI-$NO_2$ and CO and in DU for $O_3$

| Parameter | Month | MOD | SAT | MB | MAE | R | RMSE |
|-----------|-------|-----|-----|-----|-----|-----|------|
| $NO_2$ ($x10^{15}$) | June | 1.6 | 1.1 | 0.50 | 0.74 | 0.59 | 0.16 |
| | Dec | 1.2 | 1.0 | 0.18 | 0.54 | 0.58 | 0.15 |
| $O_3$ | | 39.6 | 19.3 | 20.0 | 20.0 | 0.53 | 1.70 |
| | | 33.1 | 17.3 | 15.4 | 15.4 | 0.69 | 1.62 |
| CO ($x10^{18}$) | | 1.93 | 1.92 | 0.02 | 0.03 | 0.82 | 0.03 |
| | | 1.91 | 1.79 | 0.12 | 0.12 | 0.40 | 0.04 |
| AOD | | 0.85 | 0.54 | 0.3 | 0.32 | 0.65 | 0.22 |
| | | 0.28 | 0.28 | 0.0 | 0.11 | 0.30 | 0.13 |

## 4.3 Model performance with respect to AOD

### 4.3.1 AERONET

The analysis of daily mean AOD at Mezaira for June 2022 (Fig. 5(a)) and DEWA for December 2022 (Fig. 5(b)) reveals the model tends to overestimate the observed AOD values, in particular in the summer month when it is the highest (Nelli et al. 2020, 2022). In June at Mezaira, the AERONET AOD shows a steady increase from around 0.5 to approximately 1.0 by the end of the month, which is in line with the expected build-up of aerosols with the annual maxima typically occurring in July (Nelli et al., 2022). The WRF-Chem model captures this upward variation but consistently overestimates the observed AOD, especially toward the end of the month. This overestimation is highlighted by the MB of 0.46. The general overestimation of the observed wind speed concerning ground-based measurements (Tables S2 and S3; Fig. S1) can at least partially explain this bias, together with an incorrect representation of the particle size distribution and hence the sedimentation rates, leading to excessive amounts of suspended dust (Ukhov et al., 2021; Parajuli et al., 2023). The moderate correlation coefficient

(r = 0.60) suggests that the model's day-to-day variability reasonably follows that observed.
This is expected, as dust lifting in the warmer months is mainly associated with the Shamal
winds (Yu et al., 2016), which are fairly well represented in the model. Conversely, at DEWA
in December (Fig. 5(b)), the observed AODs are lower, fluctuating between 0.2 and 0.3,
indicative of the season's lower aerosol concentrations (Nelli et al., 2020). The WRF-Chem
model again follows the observed variation but shows occasional significant overestimations,
most notably on December $10^{th}$, where simulated AOD spikes to 1.6, far exceeding the
observed AODs. Dust lifting in the colder months is typically associated with the passage of
mid-latitude weather systems (Nelli et al., 2022), which the WRF model does not fully
reproduce, in particular with respect to its timing (Temimi et al., 2020b; Taraphdar et al., 2021).
This discrepancy is reflected in the weak correlation coefficient (r = 0.16) and the MB of 0.05.
The overestimation of the near-surface wind speed at the location of the airport stations (Table
S2) and the WISE-UAE site (Table S3) is also in line with the higher amounts of atmospheric
dust in the model. Fig. 5 shows that, while the WRF-Chem model demonstrates the ability to
capture seasonal variations in AOD, it tends to overestimate AOD levels in both summer and
winter months, suggesting a need for calibration of the aerosol parameterization scheme in the
model or the emissions input. This comparison highlights the model's potential and limitations
in simulating the UAE-specific aerosol conditions, as well as where research is needed to
optimize the model performance.

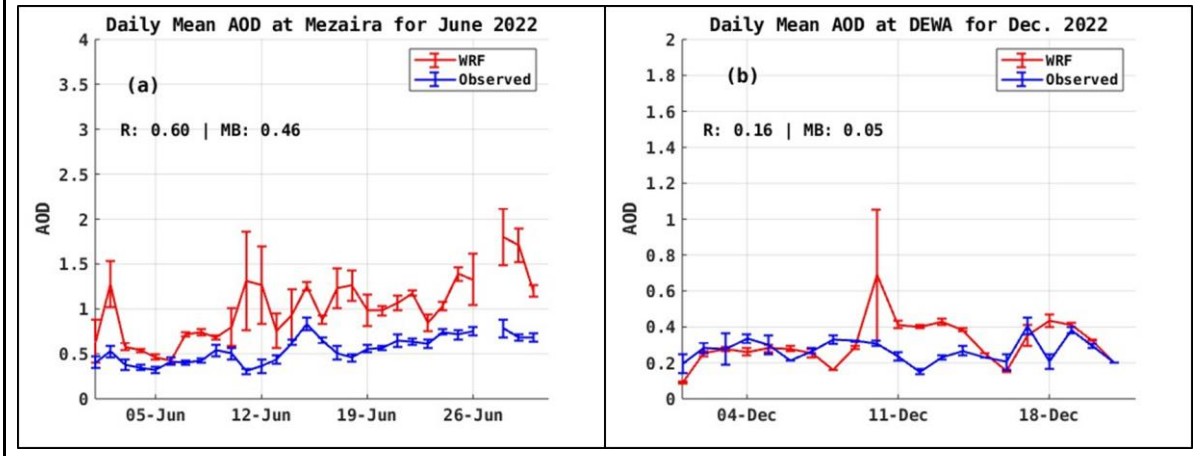

**Figure 5: Evaluation of WRF-Chem against AERONET AOD**: Daily mean Aerosol Optical Depth (AOD; dimensionless) from WRF-Chem simulations (red) and AERONET observations (blue) at Mezaira during June 2022 (a) and Dewa during December 2022 (b). The lines give the daily mean values and the error bars show one standard deviation from the

mean computed using the hourly values. The correlation (r) and mean bias (MB) are given in the plot.


## 4.3.2 MODIS

The comparison between WRF-Chem simulated and MODIS AOD (MOD-AOD) is
depicted in Fig. 6, with the statistical comparisons summarized in Table 2. The satellite-derived
MOD-AOD values follow the same seasonal cycle as the ground-based AERONET
observations: they are higher in the summer, averaging 0.54, and lower in winter, averaging
0.28, reflecting the annual cycle in aerosol loading in the region (Nelli et al., 2020). The WRF-
Chem simulations capture these seasonal variations, with corresponding AODs of 0.85 in
summer and 0.28 in winter. The model AOD demonstrates moderate correlation with MODIS
AOD, yielding correlation coefficients of 0.65 for summer and 0.30 for winter, similar to the
ones with respect to the AERONET AOD, indicating the satellite-derived and ground-based
AOD estimates are in close agreement, which has been noted by Nelli et al. (2020). The WRF-
Chem model systematically overestimates AOD by 0.31 in summer, a similar (albeit of a
smaller magnitude) bias with that respect to the AERONET station (Fig. 5(a)), while slightly
underestimating by 0.004 in winter.

For June, WRF-Chem generally overestimates AOD compared to the MODIS' estimates, in
particular over the southern and central UAE, as shown in the spatial distribution of the
difference between them (Figs. 6(a)-(c)). The frequency distribution shows most differences
clustering around zero, with a slight positive skew, reinforcing the model's overestimation
tendency for this month. Stronger wind speeds and an incorrect representation of the dust
physical and optical properties can explain the model bias. In contrast, in December there are
more balanced results, with WRF-Chem showing a closer alignment with MODIS AOD on
average. The spatial distribution of the model bias displays areas in the central and southern
UAE where the MODIS AOD exceeds the WRF-Chem values, with anomalies of the opposite
sign over the Arabian Gulf and parts of the Al Hajar mountains in Oman. Mostamandi et al.
(2023) showed that, over the Arabian Gulf and in the WRF-Chem model, the dust deposition
rates decrease away from the coastlines, with coastal UAE having lower deposition rates than
inland sites. Excessive dust deposition over the Rub Al Khali Desert is consistent with a clearer
atmosphere closer to the coastlines in the model when compared to the MODIS measurements.

The positive bias over the Arabian Gulf can be attributed to higher amounts of dust transported upstream by north-westerly winds and/or reduced dust deposition over the water in WRF-Chem. The frequency distribution in December shows a balanced spread around zero, suggesting a more accurate seasonal fit than in June. These findings, together with those in Fig. 5 with respect to the AERONET station observations, underscore the influence of seasonal atmospheric conditions on WRF-Chem's performance and suggest the need for seasonal adjustments in the aerosol parameterization to improve model accuracy in capturing the UAE's unique aerosol dynamics.

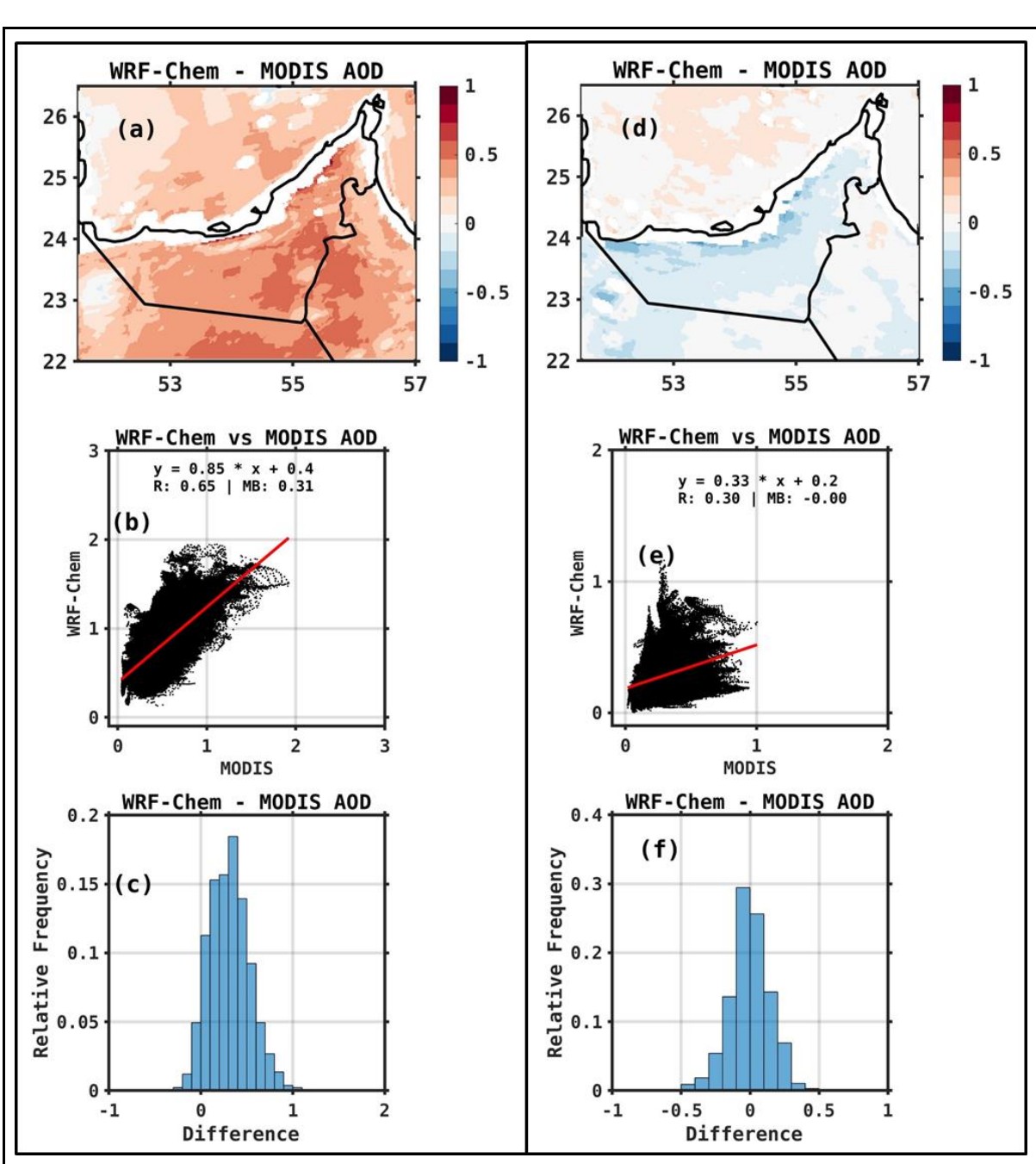

**Figure 6: Evaluation of WRF-Chem against MODIS AOD** : Same as Fig. 2 but for the MODIS AOD.

## 4.4. Aerosol influence on Ozone

Tropospheric or surface ozone ($O_3$) is one of the most significant greenhouse gases after carbon dioxide ($CO_2$) and methane ($CH_4$) (Ehhalt and Prather, 2001). It plays a critical role in the Earth's radiation budget, contributing to an increase in radiative forcing of up to 0.47 $W/m^2$ and accounting for 3-7% of global warming (Gauss, 2003; Ehhalt and Prather, 2001). Elevated $O_3$ levels in the atmospheric boundary layer are toxic and can significantly impact human health and vegetation (Adams et al., 1989). The interactions between reactive gaseous pollutants and aerosols are a major focus in the development of air quality and climate models. Aerosols, through scattering and absorption of solar radiation, influence photolysis rates and can either increase or decrease the formation of $O_3$ and its precursors (He and Carmichael, 1999). Studies have shown that aerosols impact ozone production and loss by altering photolysis frequencies (Dickerson et al., 1997; Jacobson, 1998). For example, Li et al. (2011) used an air quality model to evaluate the changes in photolysis frequencies caused by sulfate, nitrate, ammonium, and mineral dust aerosols in central and eastern China, finding a 5.4% decrease in daily average surface ozone concentrations. Similarly, Lou et al. (2014) found that aerosols reduced annual mean photolysis frequencies, $j(O^1D)$ and $j(NO_2)$, by 6–18% in polluted eastern China, resulting in reductions of up to 0.5 ppbv in $O_3$ during spring and summer, using a global chemical transport model. Attributing ozone levels to a specific source region is particularly challenging, as ozone concentrations are influenced by various processes, including stratosphere-troposphere exchange, significant hemispheric background levels, dominant local emissions, and complex photochemical reactions involving multiple trace gases (Fiore et al., 2003). Therefore, it is crucial to understand the impact of aerosol feedback on surface ozone in the UAE, a region with high aerosol loading in the Arabian Peninsula.

From Fig. 4 and the discussion in section 4.2, it is evident that ozone levels are higher during the summer season, which coincides with a dominance of aerosols over the UAE. In order to better understand the impact of aerosols on ozone concentrations, we conducted a simulation in which all aerosol components in the WRF-Chem model are turned off (No aerosol + radiative feedback on), simulating an aerosol-free atmosphere over the UAE. This simulation is

conducted alongside a control simulation (All aerosol + radiative feedback on) in which all aerosol processes are included, both for June 2022. The results of these simulations, comparing the scenarios with and without aerosols, are presented in Fig. 7 and highlight the influence of aerosols on ozone formation and spatial distribution in the region. This analysis focuses on daytime hours (04-12 UTC) and non-daytime hours (13-03 UTC) to delve deeper into ozone dynamics, as ozone production predominantly occurs during the daytime compared to non-daytime hours. Figs. 7 (a)-(b) shows the ozone distribution with and without aerosols during the daytime hours (04-12 UTC; 08-16 LT). Both panels depict higher ozone concentrations over the northern regions, with a clear gradient decreasing towards the south-eastern areas during daytime hours. The influence of aerosols on ozone production is evident in areas where the ozone levels are slightly elevated, suggesting that aerosols contribute to ozone production/loss under daytime conditions based on the nature of the aerosols. Fig. 7 (c) highlights the difference in ozone concentrations between simulations with and without aerosols for daytime hours. The difference shows localized areas of positive and negative changes, indicating regions where aerosols either enhance or suppress ozone levels. Notably, over the northern areas, particularly in oceanic regions where the ozone concentrations are the highest, the differences are generally positive, reflecting a positive feedback of aerosols on ozone production, particularly over the Arabian Gulf. On the other hand, over land areas, where the ozone is lower, the lower photolysis rates may limit ozone production. Therefore, the impact of aerosols on ozone varies based on their origin, such as dust events. These aerosols can have anthropogenic, natural, or marine origins (Filioglou et al., 2020; Nelli et al., 2021). Aerosols significantly influence surface ozone through atmospheric chemical and physical processes. Depending on their nature, aerosols can either increase or decrease ozone levels, as observed in various studies (Gao et al., 2023; Shi et al., 2022). As noted in studies such as Wang et al. (2019), Mukherjee et al. (2020), and Qu et al. (2021), the reduction in the incoming shortwave radiation flux will hinder the generation of ozone, as well as an increase in the $NO/NO_2$ ratio, which can happen when the pollutants' concentration increases in a shallower boundary layer. On the other hand, higher amounts of CO and $NO_2$ will promote the production of ozone.

Fig. 7 (d) and (e) illustrate ozone concentrations with and without aerosols for the remaining hours (non-daytime). The patterns are largely similar to those observed during the daytime, except over urban areas where the ozone concentration is much reduced owing to the lack of *in situ* generation due to the absence of sunlight and underestimation of ozone precursor concentration. Fig. 7(d) shows slightly higher concentrations than Fig. 7(e), suggesting that

aerosols continue to have an impact on ozone production, albeit less pronounced, during non-
daytime periods. Fig. 7(f) presents the difference in ozone concentrations between simulations
with and without aerosols for the non-daytime hours. The spatial distribution of positive and
negative differences follows a similar pattern to that observed during the daytime hours, though
the magnitudes are generally larger. This suggests that ozone advection from upstream sources
may play a role. Additionally, marine aerosols can contribute to ozone production through their
nature.

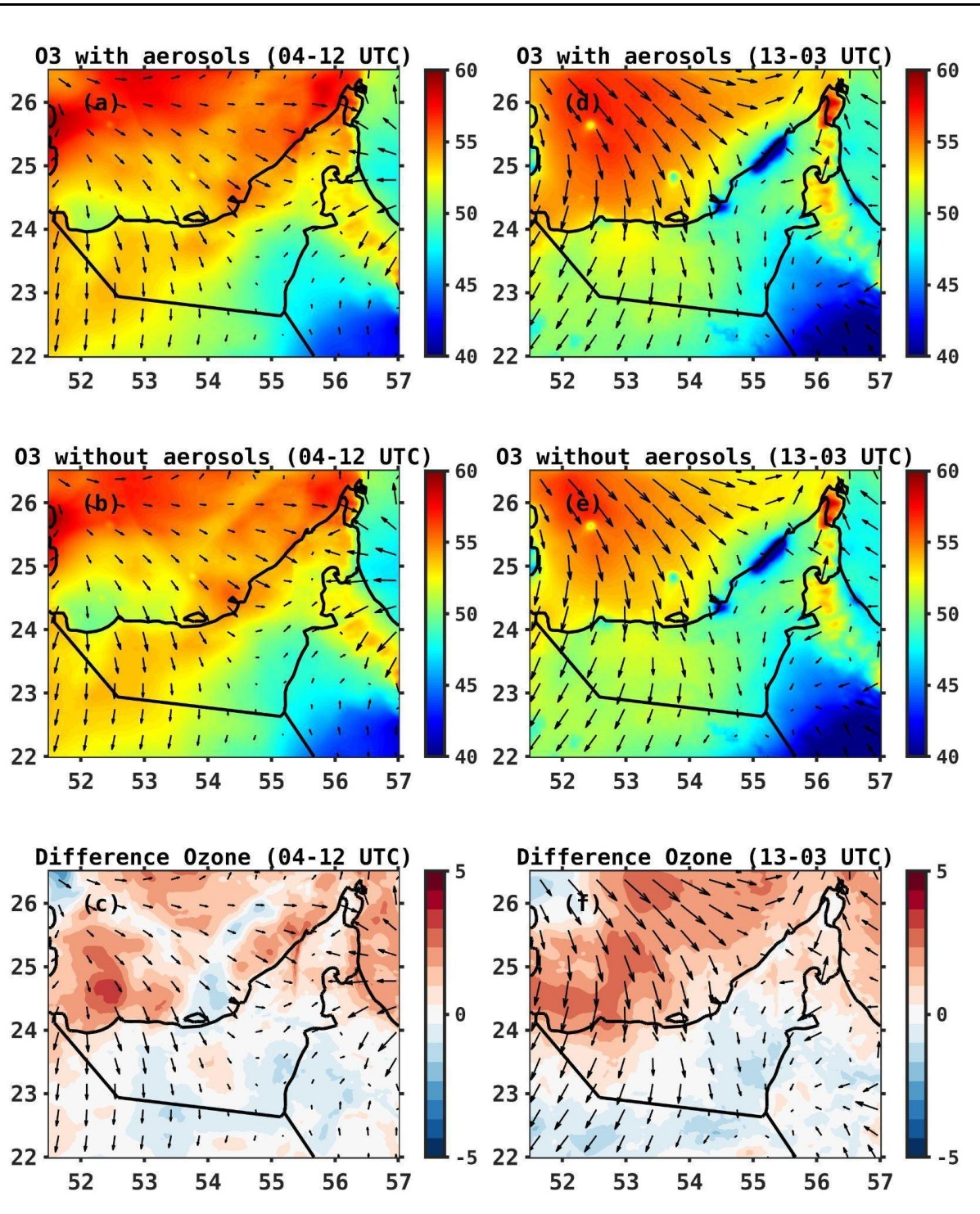

**Figure 7: Ozone (O₃) Sensitivity Simulations:** Spatial distribution of surface ozone concentrations (ppb) simulated by the WRF-Chem model with (a) and without (b) aerosols over the UAE for specified daytime hours during June 2022. (d)-(e) are as (a)-(b) for the remaining hours. Panels (c) and (f) illustrate the difference (%) in ozone concentrations (with aerosols minus without aerosols) during daytime hours and the remaining hours, respectively. The 10-m wind vectors (m/s) are overlaid on each plot, indicating the wind

patterns influencing the ozone distribution.


## 5. Conclusions

This study rigorously evaluates the performance of the Weather Research and Forecasting model coupled with chemistry (WRF-Chem) in simulating meteorological parameters and air pollutants over the United Arab Emirates (UAE) during June and December 2022, representing contrasting summer and winter conditions. The model's performance is assessed through comparisons with ground-based observations and ERA-5 reanalysis data for meteorological parameters, as well as AERONET, TROPOMI, and MODIS satellite observations for air pollutants.

We evaluated WRF-Chem model's accuracy in simulating meteorological parameters, in particularly 2-meter temperature (T2m), 10-meter wind speed (WS10m), and 2-meter relative humidity (RH2m) across 7 locations in the UAE. The model generally overestimates T2m in summer by less than 0.2 °C and underestimates it in winter by 3°C, with correlation coefficients ranging from 0.7 to 0.85 among the stations. This is comparable performance with compared to that reported studies (e.g., Branch et al., 2021; Temimi et al., 2020b), reflecting the added value of explicitly predicting chemistry fields in this aerosol-rich region. An incorrect representation of surface properties, such as the albedo and surface emissivity, and deficiencies in the model physics and dynamics, may explain the referred temperature biases. For WS10m, the model's bias is within ±0.5 m/s, indicating good agreement for both land and marine areas. The tendency for the model to overestimate the observed wind speed may arise from deficiencies in the surface drag parameterization scheme and an underrepresentation of its subgrid-scale variability (Nelli et al., 2020). In any case, and as for air temperature, the magnitude of the biases is much smaller than that reported in other studies, for which the wind speed biases exceed 3 m/s (Branch et al., 2021; Fonseca et al., 2020; Temimi et al., 2020b). The dry bias noted in these studies, however, is also seen in the WRF-Chem simulations, possibly arising from a drier soil, an incorrect representation of the mesoscale (sea-/land-breeze) circulations, and a dry bias in the forcing data. The WRF-Chem model evaluation against WISE-UAE measurements reveals a comparable performance to that seen with respect to the airport stations w.r.t T2m, WS10m and RH2m. An evaluation against the WRF-Chem values reveals the model overestimates the incoming shortwave radiation flux (SW) by about

30 W/m$^2$ for December, which can be attributed to reduced cloud cover, a known WRF deficiency (Wehbe et al., 2019; Fonseca et al. 2020, 2022a). An inspection of the diurnal cycle revealed the cold (typically by 2-3 °C) and dry (by about 20%) biases occur mostly at night, when the wind speed in the model is higher than that observed, suggesting increased advection of cooler and drier desert air into the site.

The comparison of ERA5 reanalysis data with WRF-Chem simulations revealed regional variations in T2m, specifically underestimation in the UAE's southern region and overestimation in the north-western region. Statistical metrics for summer show an underestimation of 1 °C and a correlation coefficient (r) of 0.97. In comparison, for winter a similar pattern is seen with an underestimation of 1 °C and a r value of 0.92 over the domain. The fact that WRF-Chem performs well against in-situ data and ERA5 reanalysis with respect to air temperature is also an indication that the reanalysis dataset performs well in this region, as noted by Fonseca et al. (2022b) and Nelli et al. (2024a). The mean PBL from ERA5 is largely consistent with that from the WRF-Chem outputs, with both data sets displaying a clear seasonal variation—increased PBL during summer and decreased in winter, correlating with temperature changes.

Regarding gaseous pollutants, both WRF-Chem and satellite data show higher TROPOMI-NO$_2$ columns greater than $5 \times 10^{15}$ molecules/cm$^2$ in urban and industrial regions such as Dubai, Abu Dhabi, and Ras Al Khaimah, reflecting emissions from economic activities like power generation, water desalination, and industries. Lower concentrations ($<1.5 \times 10^{15}$ molecules/cm$^2$) are noted in less urbanized areas. The WRF-Chem model closely reproduces the TROPOMI-NO$_2$ spatial patterns, even though it tends to underestimate the observed concentrations in the Abu Dhabi region and underestimate the north-eastern UAE, which has been tied to deficiencies in the emission inventory. Moderate correlation coefficients (0.59 in summer and 0.58 in winter) confirm the model's effectiveness in capturing NO$_2$'s day-to-day variability. WRF-Chem overestimates the observed TROPOMI-O$_3$ column, as indicated by positive MB values of around 11-16 DU, yet maintains high correlation coefficients (0.78 in summer and 0.83 in winter), suggesting accurate ozone concentration simulations. Colder and drier conditions, along with deficiencies in the representation of the observed chemistry, particularly concerning the NOx emissions linked to the O$_3$ concentration, can explain the WRF-Chem biases. TROPOMI-CO column simulations, on the other hand, exhibit significant discrepancies in winter and lower correlation coefficients, highlighting challenges in accurately

modelling CO levels. Besides an incorrect emission inventory, discrepancies in the representation of the atmospheric flow and its effect on the pollutant's dispersion, can explain the model performance. In summer, the analysis conducted here stresses the WRF-Chem model's strengths in simulating CO, $NO_2$ and $O_3$ columns with high fidelity with respect to the TROPOMI's observations, but also points out its limitations in estimating CO columns accurately in winter.

Regarding aerosol optical depth (AOD) observed by AERONET stations and the MODIS satellite, the WRF-Chem model generally tends to overestimate AOD, particularly during the summer months. At Mezaira in June, AERONET data showed a steady increase in AOD, which the WRF-Chem model captured but consistently overpredicted due to factors such as overestimated wind speeds and inaccuracies in particle size distribution. In December at DEWA, observed AOD levels were lower, and while the model followed the observed trends, it occasionally produced large spikes, reflecting challenges in accurately capturing the effects of mid-latitude weather systems. Correlation coefficients for AOD comparisons reveal moderate (0.60) to weak model performance depending on the season, influenced by dust transport mechanisms. Comparisons with MODIS satellite-derived AOD similarly indicated seasonal overestimations during the summer, with a closer alignment observed in winter. Spatially, overestimations in southern and central UAE in June were linked to strong winds and dust properties, while December results were more balanced. Biases over the Arabian Gulf were attributed to dust transport and deposition dynamics. Overall, the findings indicate that while the WRF-Chem model captures seasonal AOD variations, adjustments to aerosol parameterization and dust representation are necessary to improve model accuracy.

This study also explores the impact of aerosols on surface ozone ($O_3$) in the UAE by altering photolysis rates through the scattering and absorption of solar radiation. Using WRF-Chem model simulations for June 2022, we compared scenarios with and without aerosols to assess their influence. The results show higher ozone concentrations during daytime in northern regions, with aerosols contributing to localized increases or decreases. Marine aerosols notably enhance $O_3$ production over the Arabian Gulf, while lower photolysis rates limit ozone formation over land areas. During non-daytime hours, aerosol influence continues but is less significant, with urban areas experiencing reduced ozone levels due to limited photochemical activity. Additional sensitivity simulations and in-situ observations are needed to validate these findings further.

The WRF-Chem model exhibits enhanced capability in simulating key meteorological parameters and satisfactory performance in air pollutants over the UAE, showcasing significant improvements in regional-scale dynamics. This is evidenced by high skill scores with respect to observational data, with a clear improvement over previous research outcomes, particularly during summer. This comprehensive assessment validates the model's effectiveness and identifies potential areas for improvement in simulating air pollutant concentrations across the hyper-arid and aerosol-rich UAE. The discrepancies between model simulations and various observational data sets likely arise from improper emission inventories, particularly anthropogenic emissions, which must be optimized based on existing country-specific datasets. Other sources of uncertainty are model parameterization schemes and the quality of the meteorological and chemistry input data. Integrating model predictions with satellite observations and ground-based measurements is crucial for advancing air quality monitoring and enhancing the predictive accuracy of atmospheric pollutant distributions in the UAE. This collective approach aids in addressing biases and improving the overall understanding of regional air quality dynamics.

**Code and Data Availability**

The authors would like to acknowledge that the products used in this study are freely accessible online: (i) ERA-5 reanalysis data is extracted from the Copernicus Climate Change Service Climate Data Store (Hersbach et al. 2023a,b); (ii) Nitrogen Dioxide ($NO_2$), Ozone ($O_3$) and Carbon Monoxide (CO) column concentrations estimated from the measurements collected by the Tropopsheric Monitoring Instrument (TROPOMI) onboard the Sentinel 5-P satellite are extracted from the National Aeronautics and Space Administration's (NASA's) website; (iii) Aerosol Optical Depth from Moderate Resolution Imaging Spectroradiometer (MODIS) (iv) National Centers for Environmental Prediction (NCEP) Final (FNL) Operational Global Analysis meteorological data used to drive the WRF-chem simulations is downloaded from the National Center for Atmospheric Research (NCAR) Research Data Archive website (NCEP/NWS/NOAA/USDC, 2000), with the chemistry data used to force WRF-Chem, the ouput of the Community Atmosphere Model with Chemistry (CAM-chem) model, extracted from NCAR's website (Bucholz et al., 2019); (iv) the WRF-Chem model used, version 4.3.1, is freely available from the developers' website (WRF, 2023), with the pre-processor tools

available at NCAR's website (NCAR, 2023). All figures displayed in this manuscript were generated with the Matrix Laboratory (MATLAB) software version 2023 (Mathworks, 2023).

## Acknowledgment

We are thankful to the development team of the WRF-Chem model for making this model available as an open-source resource for research. We acknowledge the use of WRF-Chem pre-processor tools including mozbc, anthro_emiss, and bio_emiss provided by the Atmospheric Chemistry Observations and Modelling Lab (ACOM) of the National Center for Atmospheric Research (NCAR). Our also thanks go to the Community Atmosphere Model with Chemistry (CAM-Chem) for the chemical initial and boundary conditions. In addition, we are also thankful to the National Centers for Environmental Prediction (NCEP) Final (FNL) Operational Global Analysis data for supplying meteorological initial and lateral boundary conditions. Additionally, we are grateful to Sentinel-5P TROPOMI for satellite datasets. Finally, this research greatly benefited from the high-performance computing and research computing resources provided by Khalifa University. We express our sincere gratitude for their invaluable support. We would also like to thank the two anonymous reviewers for their several constructive and insightful comments and suggestions that helped to substantially improve the quality of this work.

## Conflict of interest

The authors declare they do not have any conflict of interest.

## Author contribution

Conceptualisation and methodology: D.F. and Y.Y.; Data curation and visualization: Y.Y.; formal analysis and interpretation: Y.Y., R.F., N.N., and D.F.; project administration and supervision: D.F.; writing—original draft: Y.Y.; review and editing: all authors.

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
