# Peer review of "Evaluation of the WRF-Chem Performance for the air pollutants over the United Arab Emirates"

_EGUsphere, 2024_

## Author Comment (AC1)

**EGUSPHERE-2024-959 | Research article**

Submitted on 29 Mar 2024

**Evaluation of the WRF-Chem Performance for gaseous pollutants over the United Arab Emirates**

Yesobu Yarragunta, Diana Francis, Ricardo Fonseca, and Narendra Nelli

Handling editor: Christoph Gerbig, cgerbig@bgc-jena.mpg.de

**Reviewer 1**

**Reviewer:** The paper presents high-resolution WRF-Chem model simulations for the UAE region. The authors carried out an evaluation of the model simulations for different meteorological and chemical fields by using ground and satellite-based observations. Given the rapid industrial development and urbanization in the region, it's important to develop state-of-the-art air quality models to simulate air pollution in UAE. The authors conducted comprehensive WRF-Chem model simulations with 3 nested grids centered over the UAE. There are a number of shortcomings of this study that need to be addressed before considering this manuscript for publication.

Reply: Thank you for your detailed review and valuable feedback on our manuscript. We appreciate the time and effort you have invested in evaluating our work. We acknowledge the importance of addressing the shortcomings you highlighted to enhance the quality and robustness of our study. Below, we outline our responses and the actions we plan to take to address each of the concerns raised:

**Major comments:**

**Reviewer:** The manuscript can be shortened quite a bit. There's plenty of text devoted to the presentation of the evaluation of the meteorological simulations by the WRF-Chem model. Although the main scope of the paper is the evaluation of the gaseous pollutants, there's lengthy discussion about the meteorological simulations in the main text. The discussion about the ERA-

5 reanalysis, evaluation of the meteorological simulations using the weather observations and ERA5 reanalysis data should be moved to the supplemental material.

**Reply:** Thank you for your insightful feedback on our manuscript. We appreciate your suggestions on how to streamline the presentation of our work. We understand that the manuscript can be shortened by focusing more on evaluating air pollutants and less on the meteorological parameters. In response to the reviewer's comment, we have taken the following actions to rearrange the manuscript: (i) we have significantly shortened the text devoted to presenting the meteorological parameters by the WRF-Chem model. This ensures that the main focus remains on evaluating the gaseous pollutants, which is the primary scope of our study; (ii) the detailed discussion about evaluating meteorological simulations using weather observations and ERA5 reanalysis data has been moved to supplemental material, as suggested. This includes comprehensive tables, figures, and statistical analyses to support the validity of the meteorological simulations, allowing interested readers to access this information without detracting from the main narrative of the paper. By implementing these changes, we have enhanced the clarity and focus of our manuscript while still providing a thorough evaluation of our meteorological simulations for those who seek it. We appreciate your guidance and thank you again for your valuable feedback.

**Reviewer:** The EDGAR emission inventory doesn't provide information on the day-to-day (e.g. weekday vs. weekend) and hourly variability (diurnal cycle) of the anthropogenic emissions. This deficiency of the emission inventory and its impact on the findings of the modeling study aren't discussed.

**Reply**: Thank you for your constructive feedback regarding using the EDGAR emission inventory in our modeling study. We appreciate your attention to detail and your suggestions for improving our manuscript.

We acknowledge that the EDGAR emission inventory does account for day-to-day (e.g., weekday vs. weekend) and hourly variability (diurnal cycle) of anthropogenic emissions, as discussed in the article by Crippa et al. (2020). To improve our manuscript, we have re-run the model simulations incorporating the diurnal variability of emissions during recent years. We have included an idealized diurnal profile for various air pollutants, assessed the impact of this variability, and presented the findings in our revised manuscript. Although EDGAR includes temporal variability, it may not capture the full complexity needed for our modeling study. We have elaborated on these limitations, emphasizing the challenges in accurately representing day-to-day and hourly emission variations (lines xx-xx). In the text, we have analyzed and discussed the potential impact of these limitations on our modeling results. Specifically, we explore how the lack of fine-grained temporal variability in the emissions might influence the accuracy and reliability of simulated air quality patterns in lines xx-xx. This is particularly relevant for short-term and diurnal analyses.

We have included a portion that explicitly addresses the limitations of the EDGAR emission inventory concerning temporal variability. This will help readers understand our modeling study's

potential shortcomings and context. We have incorporated an analysis discussing how these limitations might affect our findings, providing a nuanced view of the model's strengths and weaknesses. In the end, we outline practical steps for future research to enhance the temporal resolution of emission data and improve model accuracy. We appreciate your valuable feedback and will make these revisions accordingly. Thank you once again for your thoughtful review.

**Reviewer:** How are the emissions from point sources (e.g. power plants) ingested into the model? Is their vertical distribution taken into account?

**Reply:** Thank you for your valuable question. Our current simulations focused primarily on surface air quality; therefore, the vertical distribution of emissions from point sources, such as power plants, is not included, even though their spatial distribution is accounted for as given by the EDGAR dataset. However, we recognize the importance of accounting for the vertical distribution of emissions, especially since significant sources, such as combustion stacks, release pollutants at elevated altitudes. In future model simulations, we plan to incorporate vertical emission profiles to better represent these elevated emissions, which we note in the text (lines xx-xx). As the reviewer suggested, using vertical emission profiles, similar to temporal and speciation profiles, would provide a more accurate representation of the dispersion and impact of such emissions. In the present study, we chose to simplify the model setup by injecting all emissions into the lowest model layer. We appreciate your suggestion and will consider incorporating vertical emission profiles in future work to improve model accuracy. Thank you again for your insightful feedback.

**Reviewer:** The main weakness of this study is the lack of the model evaluation against ground-based air quality observations. Without such model evaluation it's hard to determine the applicability of the model to air quality research and prediction applications. I assume there are surface based $O_3$ and PM2.5 monitoring sites available in UAE and neighboring countries, which could be utilized.

**Reply:** Thank you for your insightful feedback. We agree that model evaluation against ground-based air quality observations is crucial for assessing the model's applicability to air quality research and prediction. Unfortunately, we cannot access ground-based air quality observations in the UAE and neighboring regions. Despite this limitation, we have aimed to provide a comprehensive analysis using the data sets noted in section 3. We will continue to seek opportunities to obtain ground-based observations in future studies to enhance the evaluation and validation of our model results.

**Reviewer:** AOD verification by using the available data from the AERONET sites located in the region would be very helpful as well.

**Reply:** Thank you for your valuable suggestion. We agree that evaluating the model's AOD using data from the available AERONET sites in the region would significantly enhance the validation

of our model results. The revised manuscript incorporates AOD data from these sites to compare with our WRF-Chem simulations for this region. This analysis will improve the robustness of our findings and contribute to the overall quality of the paper. Thank you for this insightful recommendation.

**Reviewer:** The model evaluation against the satellite observations of CO, $O_3$ and $NO_2$ is worth of publication. It'd be helpful to add information on uncertainties associated with these observations. But again the satellite observations are sporadic and can't provide information on vertical distribution of the pollutants.

**Reply**: Thank you for your valuable feedback. We have addressed the uncertainties associated with each satellite observation in the manuscript's Data Sets and Methodology section (Section 3.3). Specifically, the uncertainties for $NO_2$ observations are discussed in lines 268-270, for CO in lines 291-298, and for $O_3$ in lines 309-311. While satellite observations provide crucial data, we acknowledge their limitations, such as sporadic coverage and the lack of information on the vertical distribution of pollutants, as mentioned by the reviewer. Given this, we focused on tropospheric column observations of these pollutants. Additionally, we ensured that at least one observation is available daily, corresponding to a satellite crossing the equator at a local solar time of 13:30, to generate robust statistics. This is noted in the text (lines 241-242).

**Reviewer:** Did you include a dust parameterization in the model? Given how significant the dust emissions in the region, it's important to discuss the role of the dust aerosols and their impact on photolysis, thus affecting $O_3$ chemistry.

**Reply**: Thank you for your valuable feedback. Our model simulations employed the Model for Ozone and Related Chemical Tracers (MOZART) chemical mechanism for gas-phase species and the Georgia Institute of Technology–Goddard Global Ozone Chemistry Aerosol Radiation and Transport (GOCART) aerosol scheme to account for dust emissions. The GOCART dust emission scheme is a widely used parameterization in atmospheric models for simulating dust emissions, including the Arabian Peninsula, where it performs well (e.g. Karamuri et al., 2024). We will cite the relevant articles in the revised manuscript. This scheme calculates dust emissions based on several critical factors, including near-surface wind speed, soil moisture, surface type (such as vegetation cover and roughness length), and soil erodibility. The GOCART scheme also categorizes dust particles into bins of multiple sizes to represent a range of particle sizes, from fine dust to larger particles. These dust particles can interact with shortwave and longwave radiation fluxes influencing radiative heating rates (direct and semi-direct effects) as well as with cloud microphysics by acting as cloud condensation nuclei (CCN) or ice nuclei (IN), with indirect effects.

We recognize the significant role of dust aerosols in the region and their potential impact on photolysis rates, which can influence ozone ($O_3$) chemistry. To address this, we will conduct additional model simulations focusing on a particular dust storm event during the study period.

These simulations will analyze the impact of dust aerosols on photolysis rates and $O_3$ chemistry by comparing scenarios with and without the inclusion of dust aerosol schemes. This approach will provide deeper insights into the effect of dust aerosols on $O_3$ chemistry, especially in regions with high aerosol loads. We will also include a more detailed discussion in the revised manuscript highlighting the role of dust aerosols in altering photolysis rates and their subsequent effect on $O_3$ chemistry. We appreciate your insightful comments and believe this additional analysis will strengthen the manuscript. Thank you again for your feedback.

**Reviewer:** Was the aerosol feedback turned on in the model?

**Reply:** Thank you for your question. Yes, the aerosol-radiation and aerosol-cloud feedbacks were activated in our WRF-Chem model simulations. Given the high aerosol loads in the study region, it was essential to account for the interaction between aerosols and radiation/cloud microphysics to accurately capture their effects on atmospheric processes. Thank you for your attention to this detail.

**Reviewer:** The transport of the pollutants from other countries to UAE should be considered here as well. It'd be helpful to conduct sensitivity simulations to estimate the impact of the transboundary pollution in the region.

**Reply**: Thank you for your valuable suggestion to include sensitivity simulations to estimate the impact of transboundary pollution from other countries to the UAE. We acknowledge the importance of understanding the transboundary pollution's role in regional air quality and plan to incorporate such simulations in future studies. The current study primarily focuses on evaluating chemical and meteorological parameters within the UAE, and it represents the first effort to establish WRF-Chem simulations in this area, which is characterized by significant industrial emissions and high pollution events. As stated at the end of the manuscript, when alluding to future lines of research, a thorough assessment of the contribution of pollutants emitted from upwind regions to the air quality in the UAE has to be conducted. It will have significant implications for the design of mitigation and adaptation policies aiming at improving the air quality in the country. We hope the reviewer understands our decision.

**Reviewer:** Line 614: Are there any $NO_2$ emissions included? Usually 8-10% of NO is emitted as $NO_2$.

**Reply:** Thank you for pointing this out. Yes, our model simulations include NOx emissions from the EDGAR inventory. In our setup, we assign 90% of the NOx emissions as NO and 10% as $NO_2$, following the typical emission ratio in the region as documented e.g. in Habeebullah et al. (2015). This approach ensures that NO and $NO_2$ are appropriately represented in the model, consistent with the standard practices where 8-10% of NOx is emitted as $NO_2$. We appreciate your attention to this detail and will ensure that this is clearly stated in the revised manuscript. Thank you again for your valuable feedback.

**Reviewer:** 643-645: How was the nocturnal mixing of the chemical species parameterized in the model? WRF-Chem applies enhanced mixing within the areas with high anthropogenic emissions.

**Reply**: Thank you for your insightful comment. In our WRF-Chem model simulations, the nocturnal mixing of chemical species was parameterized using the Yonsei University (YSU) planetary boundary layer (PBL) scheme. This scheme effectively captures shallower and more stable nocturnal conditions by reducing turbulence and mixing at night, which is essential for accurately representing the vertical distribution and concentration gradients of pollutants, including $NO_x$ and VOCs, during night-time (Hoshyaripour et al., 2016). We acknowledge that enhanced nocturnal mixing, especially in urban areas with high anthropogenic emissions, can lead to variations in the model's representation of atmospheric chemistry. Badia & Jorba (2015) found that overestimating the OH radical in the model could suggest an overly oxidized atmosphere, potentially influencing nocturnal chemistry and accumulating chemical species in the surface layers. In light of the reviewer's comment, we will provide additional clarification in the revised manuscript regarding the parameterization of nocturnal mixing and its implications for chemical species distribution in our model setup. Thank you again for your valuable feedback.

**Reviewer:** 687: Are you referring to dry deposition of CO? This part needs more clarification.

**Reply**: Thank you for your comment. Yes, we are referring to the dry deposition of CO. As Kumar et al. (2022) highlighted, the absence of a vertical distribution of industrial emissions in the model can result in a rapid deposition of CO at the surface. We will clarify this point in the revised manuscript to provide a better understanding of the deposition processes considered in our simulations.

**Reviewer:** 721: This interpretation is vague.

**Reply**: Thank you for your comment. We acknowledge that the interpretation in this section is not clear. We will revise this part of the text in the manuscript to provide a more straightforward and precise explanation. This update will be included in the revised manuscript.

**Reviewer:** Pages 28-29: This chapter needs to be revised quite a bit. First, some of this material is more relevant for the Introduction section. Second, there's quite a bit of textbook material describing different $NO_x$/VOC regimes affecting tropospheric $O_3$ formation. The authors don't present any sensitivity simulations to show whether $O_3$ is $NO_x$ or VOC limited in the region. There aren't either any surface $NO_x$ or VOC measurements used here to evaluate the model. Therefore, I find this discussion vague and does not point to any particular mechanism to explain the observed model biases.

**Reply:** Thank you for your comment. We acknowledge that the interpretation in this section lacks clarity and will revise it to provide a more precise and focused explanation. We also recognize that sensitivity simulations to determine whether $O_3$ is $NO_x$ or VOC limited and including surface $NO_x$

or VOC measurements would strengthen the analysis. However, such simulations require more extensive observational data, which is currently lacking in the study area. We will remove the less relevant discussion from the text and reorganize the remaining content to enhance clarity. These updates will be reflected in the revised manuscript.

**Reviewer:** 798-800: do you see this effect occurring in the model?

**Reply**: Thank you for your comment. Yes, we do observe this particular effect in the model. A comparison of WRF-simulated Sea Surface Temperatures (SSTs) with both ERA5 and Group for High-Resolution Sea Surface Temperature (GHRSST) data over the Arabian Gulf region supports this interpretation, as discussed in lines 522-538. In response to your suggestion and another reviewer's suggestion, we will include the comparison figures in the revised manuscript to provide a clearer illustration of this effect. Thank you for your suggestion.

Minor comments:

For WRF-Chem please cite this paper as well: https://journals.ametsoc.org/view/journals/bams/98/8/bams-d-15-00308.1.xml

Reply: This reference will be included in the revised manuscript.

**Reviewer:** 204: Fix "meteorology"

Reply: This modification will be included in the revised manuscript.

**References**

Badia, A., & Jorba, O. (2015). Gas-phase evaluation of the online NMMB/BSC-CTM model over Europe for 2010 in the framework of the AQMEII-Phase2 project. *Atmospheric Environment*, *115*, 657–669. https://doi.org/10.1016/j.atmosenv.2014.05.055

Crippa, M., Solazzo, E., Huang, G., Guizzardi, D., Koffi, E., Muntean, M., Schieberle, C., Friedrich, R., & Janssens-Maenhout, G. (2020). High resolution temporal profiles in the Emissions Database for Global Atmospheric Research. *Scientific Data*, *7*(1). https://doi.org/10.1038/s41597-020-0462-2

Habeebullah, T. K. (2015) Characterising $NO_2$, Its temporal variability and Association with Meteorology: A Case Study in Makkah, Saudi Arabia. EnvironmentAsia, 8, 37-44. https://doi.org/10.14456/ea.2015.21

Hoshyaripour, G., Brasseur, G., Andrade, M. F., Gavidia-Calderón, M., Bouarar, I., & Ynoue, R. Y. (2016). Prediction of ground-level ozone concentration in São Paulo, Brazil: Deterministic versus statistic models. *Atmospheric Environment*, *145*, 365–375. https://doi.org/10.1016/j.atmosenv.2016.09.061

Karamuri, R. K., Dasari, H. P., Gandham, H., Kunchala, R. K., Attada, R., Ashok, K., & Hoteit, I. (2024). Investigation of dust-induced direct radiative forcing over the Arabian Peninsula based on high-resolution WRF-Chem simulations. *Journal of Geophysical Research: Atmospheres*, *129*, e2024JD040963. https://doi.org/10.1029/2024JD040963

Kumar, R., He, C., Bhardwaj, P., Lacey, F., Buchholz, R. R., Brasseur, G. P., Joubert, W., Labuschagne, C., Kozlova, E., & Mkololo, T. (2022). Assessment of regional carbon monoxide simulations over Africa and insights into source attribution and regional transport. *Atmospheric Environment*, *277*. https://doi.org/10.1016/j.atmosenv.2022.119075

---

## Author Comment (AC2)

**EGUSPHERE-2024-959 | Research article**

Submitted on 29 Mar 2024

**Evaluation of the WRF-Chem Performance for gaseous pollutants over the United Arab Emirates**

Yesobu Yarragunta, Diana Francis, Ricardo Fonseca, and Narendra Nelli

Handling editor: Christoph Gerbig, cgerbig@bgc-jena.mpg.de

**Reviewer 2**

**Reviewer**: The paper "Evaluation of the WRF-Chem Performance for gaseous pollutants over the United Arab Emirates" by Yarragunta et al. present an evaluation of the WRF-Chem chemistry transport model implemented by the United Arab Emirates. This is done against in situ measurements for surface windspeed and temperature, another model for other meteorological variables and TROPOMI-derived satellite measurements for trace gas chemical species. While the application of the WRF-Chem model over this area has certain scientific and applicative interest, the data and methodology of comparison is clearly limited. The only objective of evaluation of a model is better fitted to other more methodological journals such as "Atmospheric Measurements and Techniques" than "Atmospheric Chemistry and Physics" in which actual geophysical results are to be presented (and this is not the case of the current manuscript).

Reply: Thank you for your valuable feedback and for emphasizing the importance of applying the WRF-Chem model in the United Arab Emirates. We understand your concern regarding the manuscript's suitability for publication in the journal "Atmospheric Chemistry and Physics". However, we strongly believe our findings offer significant insights into regional air quality dynamics, particularly in a region characterized by high aerosol loads and extreme meteorological conditions. Our study aims to enhance the scientific understanding of atmospheric processes in the hyper-arid UAE, a country representative of those in the Middle East. This is crucial for informing future research in atmospheric chemistry and physics in arid/semi-arid regions, which are projected to expand in a warming climate. The evaluation against in situ measurements and TROPOMI-derived satellite data provides a robust assessment of the model's performance, serving as a critical

foundation for further refinement and application in operational and research atmospheric studies in the Middle East and similar hyper-arid regions. We are committed to expanding the manuscript to include a more in-depth discussion of the geophysical implications of our findings and their relevance to broader atmospheric chemistry research. We believe these additions will align the manuscript more closely with the scope of "Atmospheric Chemistry and Physics" and enhance its quality so as to meet the journal's high standards. We appreciate your thoughtful review and consideration.

**Reviewer**: Moreover, the paper needs substantial major revisions to be publishable. I strongly recommend the full revision of the three major aspects:

Reply: Thank you for your comprehensive review and valuable feedback on our manuscript. We greatly appreciate your time and effort in evaluating our work. We recognize the importance of addressing the significant revisions the reviewer has suggested to improve our study's overall quality and robustness. Below, we provide our detailed responses and outline the specific actions we intend to take to address the reviewer's concerns.

**Reviewer**:  Ozone total column: The paper only evaluates ozone simulations by comparing with total column ozone retrievals from TROPOMI. The ozone total column is largely dominated by stratospheric ozone, that accounts for 90% of the total column ozone or more. The influence of tropospheric ozone in these measurements is negligible. This is not a validation of tropospheric ozone which is the only part of ozone that affects air quality, which is the aim of the paper. Stratospheric ozone is only linked with stratospheric chemistry and transport (not mentioned in the paper). Moreover, it is unclear why there is a long paragraph (lines 721-746) describing the phenomena exclusivity driving the variability of tropospheric ozone (anthropogenic precursors, NOx or COV limited photochemical regimes).

Reply: Thank you for your insightful comment. Our focus is indeed on the tropospheric column of ozone, which is directly relevant to surface air quality. We acknowledge that the total column ozone measurements are predominantly influenced by stratospheric ozone, which accounts for approximately 90% of the total column. In comparison, tropospheric ozone contributes only about 10% as stated by the reviewer. Given this, we understand that total column ozone is unsuitable for validating ground-level ozone. Due to the unavailability of TROPOMI data for June and December 2018, we have decided to refine our simulation period to more recent years, in particular June and December 2022, for which TROPOMI ozone profile data is available (product name: S5P_L2__O3__PR_HiR), allowing for a direct evaluation of the tropospheric ozone.

Furthermore, the updated simulation period aligns with the EDGAR anthropogenic emissions data availability of up to 2022. We will ensure that the revised manuscript reflects our focus on tropospheric ozone and remove any content related to stratospheric ozone that is outside the scope of our study. Thank you again for pointing out this important distinction.

**Reviewer**: This part of the paper should be fully revised. It is mandatory to include a validation of tropospheric ozone (from the surface up to the tropopause) from WRF-Chem, which is an available ozone product from TROPOMI. Also, variability of total ozone columns should be linked with stratospheric ozone and pollution-related phenomena with only tropospheric ozone.

Reply: Thank you for your valuable comments. We have carefully considered your suggestions and revised the manuscript accordingly. We have now included validation of tropospheric ozone (from the surface up to the tropopause) using WRF-Chem, leveraging the available tropospheric ozone data from TROPOMI. Additionally, we have clarified the distinction between total ozone column variability and tropospheric ozone variability in the manuscript. As the variability in the total ozone column is primarily associated with stratospheric ozone changes, we have acknowledged that this falls outside the scope of our current study. Instead, we have focused on tropospheric ozone variability directly linked to pollution-related phenomena. We have conducted simulations for the year 2022, as stated in the reply to the reviewer's previous comment, and incorporated these revisions into the relevant sections of the manuscript.

**Reviewer**: The comparison method : authors evaluate WRF-Chem by only comparing a single monthly average maps (for 2 months) for different variables, which does not consider any information on diurnal variation. This is not sufficient for a model that is expected to provide diagnostics of air quality, since air pollution outbreaks strongly vary at daily scale and they only last for a few days (1 to 10 days). This method of validation gaseous pollution should be completed with comparisons including the daily evolution (temporal evolution within the month) and it also illustrate with a comparison of the description of at least one air pollution outbreak. More in details, strong biases should be very justified (only general arguments are provided) and statistic estimators such as RMSE should be calculated again since their values are not consistent with their definition.

Reply: Thank you for your valuable comments. We understand the importance of considering diurnal variation when evaluating WRF-Chem for air quality diagnostics. Initially, our simulations did not include a diurnal component because TROPOMI satellite retrievals are available only once daily, limiting our ability to capture daily variability. However, based on the reviewer's suggestions, we are now incorporating idealized diurnal profiles into our simulations using the updated EDGAR emission inventory, available up to 2022. Additionally, we have included a detailed analysis of the temporal evolution of gaseous pollutants within the month and provide a case study of a high-pollution event to illustrate the model's performance in capturing short-lived pollution outbreaks. Furthermore, we will provide a more detailed justification for any strong biases observed and recalculate the statistical estimators, such as RMSE, to ensure they are consistent with their definitions. These updates will be reflected in the revised manuscript.

**Reviewer**: Validation of the planetary boundary height: Given that this variable is only forecasted in models or reanalysis such as ERA5, a comparison between models is not a sufficient validation. I strongly suggest adding a comparison against measurements (typically from radiosondes or

lidar). ERA5 PBL height are useful to compare its relative spatial distribution, but a validation should include absolute comparisons against measurements. It would also be important to analyze the influence of the PBL in surface air pollutant concentrations.

Reply: Thank you for your valuable feedback on our manuscript. We agree that a more comprehensive validation of the planetary boundary layer (PBL) height is necessary. While our initial comparison of the PBL height from model simulations with ERA5 data helped to understand its relative spatial distribution, we acknowledge that this approach does not provide absolute validation. Following the Reviewer's suggestion, we have incorporated comparisons against measurements, specifically using radiosonde data available twice daily at the Abu Dhabi International Airport, the only location in the UAE where such data is collected. We extracted PBL height data from WRF-Chem for the grid point nearest the airport (24.45ºN, 54.64ºE) and compared the summer and winter of 2022. We will also examine how variations in PBL height influence surface air pollutant concentrations beyond the inverse relationship between PBL depth and a given pollutant concentration seen in a daytime-nighttime comparison to better understand its impact on air quality. These revisions have been made and are reflected in the revised manuscript.

**Reviewer**: These additional minor aspects are to be revised:

Reply: Thank you for your thorough review and valuable feedback on our manuscript.

**Reviewer**: Line 318 : the definition of the AK vector should be revised; they describe the vertical sensitivity concerning the true vertical profile of the target variable in the atmosphere

Reply: Thank you for your comment. We appreciate your attention to detail. We have revised the definition of the averaging kernel (AK) vector in line 318 to accurately reflect its role in describing the vertical sensitivity to the true vertical profile of the target variable in the atmosphere. This clarification has been made to ensure the manuscript correctly represents the vertical sensitivity information provided by the AK vector. The updated definition is included in the revised manuscript.

**Reviewer**: Equation 3 : Xret seems to be related to the "retrieved variable", which is not the "model profile". Subindexes should be renamed for consistency. The same for Xtrue.

Reply: Thank you for your comment. We appreciate your careful review. In response, we have revised the notation in Equation 3 for clarity and consistency. We acknowledge that $X_{ret}$ should represent the "retrieved profile or smoothed model profile" rather than the "model profile", and $X_{true}$ should correspond to the "true model profile (raw)" of the target variable. We have updated the sub-indices throughout the manuscript to ensure consistency and accurately reflect their meanings. The clarification regarding $X_{ret}$ as the retrieved or smoothed model profile, as mentioned in line 332, has also been maintained. These changes are reflected in the revised manuscript.

**Reviewer**: Figures : each panel of all figures should have a label (a), (b), etc.. otherwise it is unclear

Reply: Thank you for your comment and your careful review. In response, we have revised all the figures in the manuscript to include labels for each panel (e.g., (a), (b), etc.) as per the reviewer's suggestion. This addition aims to improve clarity and make it easier to reference specific panels within the figures. These revisions have been incorporated into the updated manuscript.

**Reviewer**: A figure of Group for High Resolution Sea Surface Temperature can be provided. It is actually a valuable comparison against measurements. We strongly need a graphic support for the long description of this comparison in a paragraph (near line 523).

Reply: Thank you for your comment. In response to the reviewer's suggestion, we have included a figure showing the Group for High-Resolution Sea Surface Temperature (GHRSST) data in the revised manuscript. This figure provides valuable visual support for the detailed comparison discussed in the paragraph in lines 522-538. We believe this addition enhances the clarity and effectiveness of the manuscript.

**Reviewer**: Cities, locations in the figures: We need to point out at least in one map the geographical location of the cities or places described in the paragraphs.

Reply: Thank you for your comment and thorough review. In response to the reviewer's suggestion, we have revised Fig. 1b to include the geographical locations of the cities and places mentioned in the manuscript. This addition aims to enhance clarity and provide a better geographical context for the study area. We hope this revision improves the overall readability and effectiveness of the manuscript.

**Reviewer**: Lines 534-536 : we need wind vectors overlaid in the figure to understand these circulation aspects.

Reply: Thank you for your comment and thorough review. In response to the reviewer's suggestion, we have revised Figs. 3 and 4 by overlaying surface wind vectors on each plot to better illustrate the circulation aspects discussed in the manuscript. We believe this addition enhances the clarity and understanding of the figures and improves the overall readability and effectiveness of the manuscript.

**Reviewer**: Line 566: there is not "trend" between two months, but a variation. We use the term "trend" for clear multiyear evolution with many time steps.

Reply: Thank you for your comment. In response to the reviewer's suggestion, we have replaced the term "trend" with "variation" in line 566 to accurately reflect the two months' comparison. We hope this revision improves the precision and clarity of the manuscript.

**Reviewer**: Line 572 : too many digits are used to described the PBL height comparisons (e.g. 646.7 m ?) as compared to its precision.

Reply: In response to the reviewer's suggestion, we have revised the manuscript to remove excessive digits from the PBL height comparisons, ensuring that the values are presented without decimal places to better reflect their precision. We believe this revision enhances the clarity and accuracy of the manuscript.

**Reviewer**: Line 655 : It should be clearly stated that the WRF-Chem model overestimates (positive bias) by a factor of 2 both background and peaks of NO2

Reply: We have revised the manuscript to clearly state that the WRF-Chem model overestimates peaks of $NO_2$ by a factor of 2, indicating a positive bias. This overestimation is observed when comparing the average values between the TROPOMI NO2 and the model over the UAE. This clarification has been incorporated into the revised manuscript.

**Reviewer**: Line 690 : these RMSE values are not compatible with the actual differences seen visually in the maps. If it is an unbiased RMSE that is calculated, the definition and name of the quantity should be revised.

Reply: Thank you for your comment and thorough review. Our analysis used the standard RMSE computation, not the unbiased RMSE. The RMSE values in our study represent the average error across the entire UAE, which may not always correspond directly to the visual differences observed in the maps. The maps in Figs. 5, 6, and 7 show absolute differences, while the RMSE values in line 690 summarize the overall model error. We will clarify this distinction in the revised manuscript to prevent any confusion. Thank you for highlighting this point.

**Reviewer**: How are the biases, RMSE, R of WRF-Chem in other regions in literature (East Asia and India) compare to those found in this study? Provide this comparison in percentage and the comparison with the results of the paper should be explicitly stated.

Reply: Thank you for your comment and thorough review. In response to the reviewer's suggestion, we will incorporate a comparison of the biases, RMSE, and R of WRF-Chem from other geographical regions, such as East Asia and India, as reported in the literature. This comparison will be presented as a percentage and explicitly discussed concerning the results of our study. We believe this addition will enhance the clarity and comprehensiveness of the manuscript.

**Reviewer**: Ozone columns are often express in Dobson Units. For comparison, this unit should be used.

Reply: Thank you for your comment. In response to the reviewer's suggestion, we have revised the manuscript to express the ozone columns in Dobson Units (DU) for consistency and ease of

comparison. We believe this modification will improve the clarity and comprehensiveness of the manuscript.

**Reviewer**: What is the influence of the altitude in the comparison between gaseous pollutants ? Where do averaging kernels peak? Model values without AVK should be shown as well to understand its influence.

Reply: Thank you for your comment. In response to the reviewer's suggestion, we have revised the manuscript to include the raw model values in the corresponding tables to understand better the influence of averaging kernels on comparing gaseous pollutants. Additionally, we have incorporated a figure showing the averaging kernel matrix to illustrate where the averaging kernels peak in the revised manuscript. These revisions have been made to enhance the clarity and comprehensiveness of the manuscript.

---

## Author Response (AR1)

**Reviewer 1**

**Reviewer:** The paper presents high-resolution WRF-Chem model simulations for the UAE region. The authors carried out an evaluation of the model simulations for different meteorological and chemical fields by using ground and satellite-based observations. Given the rapid industrial development and urbanization in the region, it's important to develop state-of-the-art air quality models to simulate air pollution in UAE. The authors conducted comprehensive WRF-Chem model simulations with 3 nested grids centered over the UAE. There are a number of shortcomings of this study that need to be addressed before considering this manuscript for publication.

Reply: Thank you for your detailed review and valuable feedback on our manuscript. We appreciate the time and effort you have invested in evaluating our work. We acknowledge the importance of addressing the shortcomings you highlighted to enhance the quality and robustness of our study. Below, we outline our responses and the actions we plan to take to address each of the concerns raised:

**Major comments:**

**Reviewer:** The manuscript can be shortened quite a bit. There's plenty of text devoted to the presentation of the evaluation of the meteorological simulations by the WRF-Chem model. Although the main scope of the paper is the evaluation of the gaseous pollutants, there's lengthy discussion about the meteorological simulations in the main text. The discussion about the ERA-5 reanalysis, evaluation of the meteorological simulations using the weather observations and ERA5 reanalysis data should be moved to the supplemental material.

**Reply:** Thank you for your insightful feedback on our manuscript. We appreciate your suggestions on how to streamline the presentation of our work. We understand that the manuscript can be shortened by focusing more on evaluating air pollutants and less on the meteorological parameters. In response to the reviewer's comment, we have taken the following actions to rearrange the manuscript: (i) we have significantly shortened the text devoted to presenting the meteorological parameters by the WRF-Chem model (lines 469-512). This ensures that the main focus remains on evaluating the gaseous pollutants, which is the primary scope of our study; (ii) the detailed discussion about evaluating meteorological simulations using weather observations and ERA5 reanalysis data has been moved to supplemental material, as suggested. This includes comprehensive tables, figures, and statistical analyses to support the validity of the meteorological simulations, allowing interested readers to access this information without detracting from the main narrative of the paper. By implementing these changes, we have enhanced the clarity and focus of our manuscript while still providing a thorough evaluation of our meteorological simulations for those who seek it (lines 469-533). We appreciate your guidance and thank you again for your valuable feedback.

**Reviewer:** The EDGAR emission inventory doesn't provide information on the day-to-day (e.g. weekday vs. weekend) and hourly variability (diurnal cycle) of the anthropogenic emissions. This deficiency of the emission inventory and its impact on the findings of the modeling study aren't discussed.

**Reply**: Thank you for your constructive feedback regarding using the EDGAR emission inventory in our modeling study. We appreciate your attention to detail and your suggestions for improving our manuscript.

We acknowledge that the EDGAR emission inventory does account for day-to-day (e.g., weekday vs. weekend) and hourly variability (diurnal cycle) of anthropogenic emissions, as discussed in the article by Crippa et al. (2020). To improve our manuscript, we have re-run the model simulations incorporating the diurnal variability of emissions during recent years. We have included an idealized diurnal profile for various air pollutants, assessed the impact of this variability, and presented the findings in our revised manuscript. Although EDGAR includes temporal variability, it may not capture the full complexity needed for our modeling study. We have elaborated on these limitations, emphasizing the challenges in accurately representing day-to-day and hourly emission variations (lines 218-238). In the text, we have analyzed and discussed the potential impact of these limitations on our modeling results. Specifically, we explore how the lack of fine-grained temporal variability in the emissions might influence the accuracy and reliability of simulated air quality patterns.This is particularly relevant for short-term and diurnal analyses.

**Reviewer:** How are the emissions from point sources (e.g. power plants) ingested into the model? Is their vertical distribution taken into account?

**Reply:** Thank you for your valuable question. Our current simulations focused primarily on surface air quality; therefore, the vertical distribution of emissions from point sources, such as power plants, is not included, even though their spatial distribution is accounted for as given by the EDGAR dataset. However, we recognize the importance of accounting for the vertical distribution of emissions, especially since significant sources, such as combustion stacks, release pollutants at elevated altitudes. In future model simulations, we plan to incorporate vertical emission profiles to better represent these elevated emissions, which we note in the text (lines 238-247). As the reviewer suggested, using vertical emission profiles, similar to temporal and speciation profiles, would provide a more accurate representation of the dispersion and impact of such emissions. In the present study, we chose to simplify the model setup by injecting all emissions into the lowest model layer. We appreciate your suggestion and will consider incorporating vertical emission profiles in future work to improve model accuracy. Thank you again for your insightful feedback.

**Reviewer:** The main weakness of this study is the lack of the model evaluation against ground-based air quality observations. Without such model evaluation it's hard to determine the applicability of the model to air quality research and prediction applications. I assume there are

surface based $O_3$ and PM2.5 monitoring sites available in UAE and neighboring countries, which could be utilized.

**Reply:** Thank you for your insightful feedback. We agree that model evaluation against ground-based air quality observations is crucial for assessing the model's applicability to air quality research and prediction. Unfortunately, we cannot access ground-based air quality observations in the UAE and neighboring regions. Despite this limitation, we have aimed to provide a comprehensive analysis using the data sets noted in section 3. We will continue to seek opportunities to obtain ground-based observations in future studies to enhance the evaluation and validation of our model results.

**Reviewer:** AOD verification by using the available data from the AERONET sites located in the region would be very helpful as well.

**Reply:** Thank you for your valuable suggestion. We agree that evaluating the model's AOD using data from the available AERONET sites in the region would significantly enhance the validation of our model results. The revised manuscript incorporates AOD data from available sites to compare with our WRF-Chem simulations for this region. This analysis will improve the robustness of our findings and contribute to the overall quality of the paper. Thank you for this insightful recommendation. The available AERONET sites are discussed in Section 3.2 (lines 276-285), and the subsequent evaluation results are presented in Section 4.3.1 (lines 726-759)

**Reviewer:** The model evaluation against the satellite observations of CO, $O_3$ and $NO_2$ is worth of publication. It'd be helpful to add information on uncertainties associated with these observations. But again the satellite observations are sporadic and can't provide information on vertical distribution of the pollutants.

**Reply**: Thank you for your valuable feedback. We have addressed the uncertainties associated with each satellite observation in the manuscript's Data Sets and Methodology section (Section 3.4). Specifically, the uncertainties for $NO_2$ observations are discussed in lines 324-332, for CO in lines 347-353, and for $O_3$ in lines 363-365. While satellite observations provide crucial data, we acknowledge their limitations, such as sporadic coverage and the lack of information on the vertical distribution of pollutants, as mentioned by the reviewer. Given this, we focused on tropospheric column observations for NO2 and O3. In contrast, the total column observations for CO. Additionally, we ensured that at least one observation is available daily, corresponding to a satellite crossing the equator at a local solar time of 13:30, to generate robust statistics.

**Reviewer:** Did you include a dust parameterization in the model? Given how significant the dust emissions in the region, it's important to discuss the role of the dust aerosols and their impact on photolysis, thus affecting $O_3$ chemistry.

**Reply**: Thank you for your valuable feedback. Our model simulations employed the Model for Ozone and Related Chemical Tracers (MOZART) chemical mechanism for gas-phase species and the Georgia Institute of Technology–Goddard Global Ozone Chemistry Aerosol Radiation and Transport (GOCART) aerosol scheme to account for dust emissions (Section 2). The GOCART dust emission scheme is a widely used parameterization in atmospheric models for simulating dust emissions, including the Arabian Peninsula, where it performs well (e.g. Karamuri et al., 2024). We have cited the relevant articles in the revised manuscript. This scheme calculates dust emissions based on several critical factors, including near-surface wind speed, soil moisture, surface type (such as vegetation cover and roughness length), and soil erodibility. The GOCART scheme also categorizes dust particles into bins of multiple sizes to represent a range of particle sizes, from fine dust to larger particles. These dust particles can interact with shortwave and longwave radiation fluxes influencing radiative heating rates (direct and semi-direct effects) as well as with cloud microphysics by acting as cloud condensation nuclei (CCN) or ice nuclei (IN), with indirect effects.

We recognize the significant role of dust aerosols in the region and their impact on photolysis rates, which influence ozone ($O_3$) chemistry. To address this, we conducted additional model simulations focused on the summer period, during which no significant dust storms were observed. These simulations compared scenarios with and without aerosol inclusion to analyze their impact on photolysis rates and $O_3$ chemistry, offering deeper insights into the influence of aerosols on ozone levels.

Section 4.4 explores this analysis, highlighting how dust and sea salt aerosols can enhance or suppress ozone levels. Additionally, we assessed the analysis during daytime and non-daytime hours, noting that ozone concentrations are generally higher during the day. The findings indicate that aerosols impact ozone similarly during these periods, with significant deviations observed. This is included in the revised manuscript in lines 799-868. This study represents the first effort to use WRF-Chem simulations to evaluate chemical parameters in the UAE. As noted in the conclusion, future research should comprehensively assess the contributions of different aerosols to air quality in the region. We appreciate the reviewer's insightful comments, which have strengthened the manuscript. Thank you for your feedback.

**Reviewer:** Was the aerosol feedback turned on in the model?

**Reply:** Thank you for your question. Yes, the aerosol-radiation and aerosol-cloud feedbacks were activated in our WRF-Chem model simulations. Given the high aerosol loads in the study region, it was essential to account for the interaction between aerosols and radiation/cloud microphysics to capture their effects on atmospheric processes accurately. Thank you for your attention to this detail.

**Reviewer:** The transport of the pollutants from other countries to UAE should be considered here as well. It'd be helpful to conduct sensitivity simulations to estimate the impact of the transboundary pollution in the region.

**Reply**: Thank you for your valuable suggestion to include sensitivity simulations to estimate the impact of transboundary pollution from other countries to the UAE. We acknowledge the importance of understanding transboundary pollution's role in regional air quality and plan to incorporate such simulations in future studies. The current study primarily focuses on evaluating chemical parameters within the UAE, and it represents the first effort to establish WRF-Chem simulations in this area, which is characterized by significant industrial emissions and high pollution events. As stated at the end of the manuscript, when alluding to future lines of research, a thorough assessment of the contribution of pollutants emitted from upwind regions to the air quality in the UAE has to be conducted. It will have significant implications for designing mitigation and adaptation policies aiming to improve the country's air quality. We hope the reviewer understands our decision.

**Reviewer:** Line 614: Are there any $NO_2$ emissions included? Usually 8-10% of NO is emitted as $NO_2$.

**Reply:** Thank you for pointing this out. Yes, our model simulations include NOx emissions from the EDGAR inventory for 2022. In our setup, we assign 90% of the NOx emissions as NO and 10% as $NO_2$, following the typical emission ratio in the other arid regions as documented, e.g., in Habeebullah et al. (2015). This approach ensures that NO and $NO_2$ are appropriately represented in the model, consistent with the standard practices where 8-10% of NOx is emitted as $NO_2$. We appreciate your attention to this detail (lines 548-549). Thank you again for your valuable feedback.

**Reviewer:** 643-645: How was the nocturnal mixing of the chemical species parameterized in the model? WRF-Chem applies enhanced mixing within the areas with high anthropogenic emissions.

**Reply**: Thank you for your insightful comment. In our WRF-Chem model simulations, the nocturnal mixing of chemical species was parameterized using the Yonsei University (YSU) planetary boundary layer (PBL) scheme. This scheme effectively captures shallower and more stable nocturnal conditions by reducing turbulence and mixing at night, which is essential for accurately representing the vertical distribution and concentration gradients of pollutants, including $NO_x$ and VOCs, during night-time (Hoshyaripour et al., 2016). We acknowledge that enhanced nocturnal mixing, especially in urban areas with high anthropogenic emissions, can lead to variations in the model's representation of atmospheric chemistry. Badia & Jorba (2015) found that overestimating the OH radical in the model could suggest an overly oxidized atmosphere, potentially influencing nocturnal chemistry and accumulating chemical species in the surface layers (lines 571-579). Thank you again for your valuable feedback.

**Reviewer:** 687: Are you referring to dry deposition of CO? This part needs more clarification.

**Reply**: Thank you for your comment. Yes, we are referring to the dry deposition of CO. As Kumar et al. (2022) highlighted, the absence of a vertical distribution of industrial emissions in the model can rapidly depose CO at the surface (lines 623-627).

**Reviewer:** 721: This interpretation is vague.

**Reply**: Thank you for your comment. We acknowledge that the interpretation in this section was unclear. We have revised this part of the text in the updated manuscript to provide a more straightforward and precise explanation. This revision is reflected in the revised manuscript on lines 684-700.

**Reviewer:** Pages 28-29: This chapter needs to be revised quite a bit. First, some of this material is more relevant for the Introduction section. Second, there's quite a bit of textbook material describing different $NO_x$/VOC regimes affecting tropospheric $O_3$ formation. The authors don't present any sensitivity simulations to show whether $O_3$ is $NO_x$ or VOC limited in the region. There aren't either any surface $NO_x$ or VOC measurements used here to evaluate the model. Therefore, I find this discussion vague and does not point to any particular mechanism to explain the observed model biases.

**Reply:** Thank you for your comment. We acknowledge that the interpretation in this section lacked clarity and have revised it to provide a more precise and focused explanation. We also recognize that conducting sensitivity simulations to determine whether $O_3$ formation is NOx- or VOC-limited, along with including surface NOx or VOC measurements, would strengthen the analysis. However, such simulations require more extensive observational data, which is currently unavailable for the study area. Consequently, we have removed less relevant discussions and reorganized the remaining content to enhance clarity. These updates are reflected in the revised manuscript (lines 684-700).

**Reviewer:** 798-800: do you see this effect occurring in the model?

**Reply**: Thank you for your comment. We do not observe this specific effect in our current model simulation for 2022. Cloud-related meteorological changes observed in previous model simulations appear to differ from those in the present simulation, possibly due to meteorological conditions across different simulation years.

Minor comments:

For WRF-Chem please cite this paper as well: https://journals.ametsoc.org/view/journals/bams/98/8/bams-d-15-00308.1.xml

Reply: The revised manuscript has included this reference (at line 177).

**Reviewer:** 204: Fix "meteorology"

Reply: We fixed this in the revised manuscript.

**References**

Badia, A., & Jorba, O. (2015). Gas-phase evaluation of the online NMMB/BSC-CTM model over Europe for 2010 in the framework of the AQMEII-Phase2 project. *Atmospheric Environment*, *115*, 657–669. https://doi.org/10.1016/j.atmosenv.2014.05.055

Crippa, M., Solazzo, E., Huang, G., Guizzardi, D., Koffi, E., Muntean, M., Schieberle, C., Friedrich, R., & Janssens-Maenhout, G. (2020). High resolution temporal profiles in the Emissions Database for Global Atmospheric Research. *Scientific Data*, *7*(1). https://doi.org/10.1038/s41597-020-0462-2

Habeebullah, T. K. (2015) Characterising $NO_2$, Its temporal variability and Association with Meteorology: A Case Study in Makkah, Saudi Arabia. EnvironmentAsia, 8, 37-44. https://doi.org/10.14456/ea.2015.21

Hoshyaripour, G., Brasseur, G., Andrade, M. F., Gavidia-Calderón, M., Bouarar, I., & Ynoue, R. Y. (2016). Prediction of ground-level ozone concentration in São Paulo, Brazil: Deterministic versus statistic models. *Atmospheric Environment*, *145*, 365–375. https://doi.org/10.1016/j.atmosenv.2016.09.061

Karamuri, R. K., Dasari, H. P., Gandham, H., Kunchala, R. K., Attada, R., Ashok, K., & Hoteit, I. (2024). Investigation of dust-induced direct radiative forcing over the Arabian Peninsula based on high-resolution WRF-Chem simulations. *Journal of Geophysical Research: Atmospheres*, *129*, e2024JD040963. https://doi.org/10.1029/2024JD040963

Kumar, R., He, C., Bhardwaj, P., Lacey, F., Buchholz, R. R., Brasseur, G. P., Joubert, W., Labuschagne, C., Kozlova, E., & Mkololo, T. (2022). Assessment of regional carbon monoxide simulations over Africa and insights into source attribution and regional transport. *Atmospheric Environment*, *277*. https://doi.org/10.1016/j.atmosenv.2022.119075

**Reviewer 2**

**Reviewer**: The paper "Evaluation of the WRF-Chem Performance for gaseous pollutants over the United Arab Emirates" by Yarragunta et al. present an evaluation of the WRF-Chem chemistry transport model implemented by the United Arab Emirates. This is done against in situ measurements for surface windspeed and temperature, another model for other meteorological variables and TROPOMI-derived satellite measurements for trace gas chemical species. While the application of the WRF-Chem model over this area has certain scientific and applicative interest, the data and methodology of comparison is clearly limited. The only objective of evaluation of a model is better fitted to other more methodological journals such as "Atmospheric Measurements and Techniques" than "Atmospheric Chemistry and Physics" in which actual geophysical results are to be presented (and this is not the case of the current manuscript).

Reply: Thank you for your valuable feedback and for emphasizing the importance of applying the WRF-Chem model in the United Arab Emirates. We understand your concern regarding the manuscript's suitability for publication in the journal "Atmospheric Chemistry and Physics". However, our findings offer significant insights into regional air quality dynamics, particularly in a region characterized by high aerosol loads and extreme meteorological conditions. Our study aims to enhance the scientific understanding of atmospheric processes in the hyper-arid UAE, a country representative of those in the Middle East. This is crucial for informing future atmospheric chemistry and physics research in arid/semi-arid regions, which are projected to expand in a warming climate. The evaluation against in situ measurements, TROPOMI-derived satellite data, AERONET, and MODIS satellite provides a robust assessment of the model's performance, serving as a critical foundation for further refinement and application in operational and research atmospheric studies in the Middle East and similar hyper-arid regions. We have expanded the manuscript to provide a more comprehensive discussion of the geophysical implications of our findings and their relevance to broader atmospheric chemistry research. This includes the addition of two new sections, 4.3 and 4.4. Additionally, we conducted a full simulation for the year 2022 using updated EDGAR emissions processed with an optimal diurnal profile to accurately capture pollutant variations throughout the day (lines 218-238). Section 4.3 evaluates these model simulations against MODIS-derived and AERONET-observed AOD, demonstrating moderate correlation values of 0.65 and 0.60 during the summer (lines 725-798). While the previous manuscript focused solely on gaseous pollutants and key meteorological parameters, the revised manuscript broadens its scope with an additional aerosol evaluation, enhancing its relevance to a wider research community. Furthermore, Section 4.4 is dedicated to examining the impact of aerosols on ozone levels, further strengthening the scientific contribution of this work (lines 799-868). We believe these additions can align the manuscript more closely with the scope of "Atmospheric Chemistry and Physics" and enhance its quality to meet the journal's high standards. We appreciate your thoughtful review and consideration.

**Reviewer**: Moreover, the paper needs substantial major revisions to be publishable. I strongly recommend the full revision of the three major aspects:

Reply: Thank you for your comprehensive review and valuable feedback on our manuscript. We greatly appreciate your time and effort in evaluating our work. We recognize the importance of addressing the significant revisions the reviewer has suggested to improve our study's overall quality and robustness. Below, we provide our detailed responses and outline the specific actions we intend to take to address the reviewer's concerns.

**Reviewer**: Ozone total column: The paper only evaluates ozone simulations by comparing with total column ozone retrievals from TROPOMI. The ozone total column is largely dominated by stratospheric ozone, that accounts for 90% of the total column ozone or more. The influence of tropospheric ozone in these measurements is negligible. This is not a validation of tropospheric ozone which is the only part of ozone that affects air quality, which is the aim of the paper. Stratospheric ozone is only linked with stratospheric chemistry and transport (not mentioned in the paper). Moreover, it is unclear why there is a long paragraph (lines 721-746) describing the phenomena exclusivity driving the variability of tropospheric ozone (anthropogenic precursors, NOx or COV limited photochemical regimes).

Reply: Thank you for your insightful comment. Our focus is indeed on the tropospheric column of ozone, which is directly relevant to surface air quality. We acknowledge that the total column ozone measurements are predominantly influenced by stratospheric ozone, which accounts for approximately 90% of the total column. In comparison, tropospheric ozone contributes only about 10% as stated by the reviewer. Given this, we understand that total column ozone is unsuitable for validating ground-level ozone. Due to the unavailability of TROPOMI data for June and December 2018, we have decided to refine our simulation period to more recent years, in particular June and December 2022, for which TROPOMI ozone profile data is available (product name: S5P_L2__O3__PR_HiR), allowing for a direct evaluation of the tropospheric ozone. This is stated in the revised manuscript in lines 354-371. Furthermore, the updated simulation period aligns with the EDGAR anthropogenic emissions data availability up to 2022. We acknowledge that the interpretation in this section was unclear. We have revised this part of the text in the updated manuscript to provide a more straightforward and precise explanation. This revision is reflected in the revised manuscript on lines 684-700. We also recognize that conducting sensitivity simulations to determine whether $O_3$ formation is NOx- or VOC-limited, along with including surface NOx or VOC measurements, would strengthen the analysis. However, such simulations require more extensive observational data currently unavailable for the study area. Consequently, we have removed less relevant discussions and reorganized the remaining content to enhance clarity. These updates are reflected in the revised manuscript (lines 684-700). Thank you again for pointing out this important distinction.

**Reviewer**: This part of the paper should be fully revised. It is mandatory to include a validation of tropospheric ozone (from the surface up to the tropopause) from WRF-Chem, which is an available ozone product from TROPOMI. Also, variability of total ozone columns should be linked with stratospheric ozone and pollution-related phenomena with only tropospheric ozone.

Reply: Thank you for your valuable comments. We have carefully considered your suggestions and revised the manuscript accordingly. We have now included a validation of tropospheric ozone (from the surface to the 100 hPa) using WRF-Chem, leveraging tropospheric ozone data from TROPOMI (lines 653-700). Additionally, we have clarified the distinction between total ozone column variability and tropospheric ozone variability in the revised manuscript (lines 365-371). Since changes primarily influence total ozone column variability in stratospheric ozone, we have noted that this aspect falls outside the scope of our current study. Instead, we focus on tropospheric ozone variability directly linked to pollution-related phenomena.

We conducted simulations for the year 2022, as mentioned in response to the reviewer's previous comment, and incorporated these revisions throughout the relevant sections of the manuscript. The revised manuscript focuses exclusively on tropospheric ozone, where we observed an overestimation by the model in both seasons. This variation is attributed to meteorological conditions and ozone precursor gases, as detailed in the text (lines 653-700).

**Reviewer**: The comparison method : authors evaluate WRF-Chem by only comparing a single monthly average maps (for 2 months) for different variables, which does not consider any information on diurnal variation. This is not sufficient for a model that is expected to provide diagnostics of air quality, since air pollution outbreaks strongly vary at daily scale and they only last for a few days (1 to 10 days). This method of validation gaseous pollution should be completed with comparisons including the daily evolution (temporal evolution within the month) and it also illustrate with a comparison of the description of at least one air pollution outbreak. More in details, strong biases should be very justified (only general arguments are provided) and statistic estimators such as RMSE should be calculated again since their values are not consistent with their definition.

Reply: Thank you for your valuable comments. We understand the importance of considering diurnal variation when evaluating WRF-Chem for air quality diagnostics. Initially, our simulations did not include a diurnal component because TROPOMI satellite retrievals are available only once daily, limiting our ability to capture daily variability. However, based on the reviewer's suggestions, we have incorporated idealized diurnal profiles into our simulations using the updated EDGAR emission inventory, available up to 2022 (lines 218-238).

We agree that evaluating the model against ground-based air quality observations is essential for assessing its applicability to air quality research and predictions. Unfortunately, we currently do not have access to ground-based air quality observations in the UAE and neighboring regions. However, we remain committed to seeking opportunities to obtain such data in future studies to enhance the evaluation and validation of our model results. Additionally, we have conducted a

detailed analysis of the temporal evolution of AERONET AOD, as shown in Figure 5, comparing the model simulations with AERONET station data available for June and December 2022. This detailed analysis has been incorporated into the revised manuscript (lines 726-759). We acknowledge that a comprehensive evaluation of the model's capability to diagnose air quality should include detailed daily variations and assessments of air pollution outbreaks, given their rapid changes over short timescales (1 to 10 days). However, due to current limitations, we are unable to perform such simulations and analyses within the scope of this manuscript. We agree that strong biases must be thoroughly justified and that detailed statistical estimators, including recalculated RMSE values, are important for accurate evaluation. We are committed to addressing these aspects in future studies, including focusing on air pollution outbreaks and enhancing model validation to capture daily and event-specific variability more effectively. Thank you again for your constructive feedback, and we will strive to incorporate these elements in subsequent model simulations.

**Reviewer**: Validation of the planetary boundary height: Given that this variable is only forecasted in models or reanalysis such as ERA5, a comparison between models is not a sufficient validation. I strongly suggest adding a comparison against measurements (typically from radiosondes or lidar). ERA5 PBL height are useful to compare its relative spatial distribution, but a validation should include absolute comparisons against measurements. It would also be important to analyze the influence of the PBL in surface air pollutant concentrations.

Reply: Thank you for your valuable feedback on our manuscript. We agree that a more comprehensive validation of the planetary boundary layer (PBL) height is important. While our initial comparison of PBL heights from model simulations with ERA5 data provided insights into their relative spatial distribution, we recognize that this approach does not constitute absolute validation. In the revised manuscript, we have shifted our primary focus to air pollutants rather than key meteorological parameters, as per the feedback from another reviewer. The analysis and discussion of these meteorological parameters have been moved to the supplementary material. If a more detailed evaluation of PBL height is desired, we are open to incorporating it in future revisions. Thank you for your thoughtful suggestions.

**Reviewer**: These additional minor aspects are to be revised:

Reply: Thank you for your thorough review and valuable feedback on our manuscript.

**Reviewer**: Line 318 : the definition of the AK vector should be revised; they describe the vertical sensitivity concerning the true vertical profile of the target variable in the atmosphere

Reply: Thank you for your comment. We appreciate your attention to detail. We have revised the definition of the averaging kernel (AK) vector in lines 399-401 to accurately reflect its role in describing the vertical sensitivity to the true vertical profile of the target variable in the atmosphere. The updated definition is included in the revised manuscript.

**Reviewer**: Equation 3 : Xret seems to be related to the "retrieved variable", which is not the "model profile". Subindexes should be renamed for consistency. The same for Xtrue.

Reply: Thank you for your comment. We appreciate your careful review. In response, we have revised the notation in Equation 3 for clarity and consistency. We acknowledge that $X_{ret}$ should represent the "retrieved profile or smoothed model profile" rather than the "model profile", and $X_{true}$ should correspond to the "true model profile (raw)" of the target variable. We have updated the sub-indices throughout the manuscript to ensure consistency and accurately reflect their meanings. The clarification regarding $X_{ret}$ as the retrieved or smoothed model profile, as mentioned in lines 415-419, 421-423, and 431-433, has also been maintained. These changes are reflected in the revised manuscript.

**Reviewer**: Figures : each panel of all figures should have a label (a), (b), etc.. otherwise it is unclear

Reply: Thank you for your comment and your careful review. In response, we have revised all the figures in the manuscript to include labels for each panel (e.g., Figure2 (a), (b), etc.) as per the reviewer's suggestion. This addition aims to improve clarity and make it easier to reference specific panels within the figures. These revisions have been incorporated into the updated manuscript.

**Reviewer**: A figure of Group for High Resolution Sea Surface Temperature can be provided. It is actually a valuable comparison against measurements. We strongly need a graphic support for the long description of this comparison in a paragraph (near line 523).

Reply: Thank you for your comment. In response to the reviewer's suggestion, We do not observe this specific effect in our current model simulation for 2022. Cloud-related changes in meteorology observed in previous model simulations appear to differ from those in the present simulation, possibly due to meteorological conditions across different simulation years.

**Reviewer**: Cities, locations in the figures: We need to point out at least in one map the geographical location of the cities or places described in the paragraphs.

Reply: Thank you for your comment and thorough review. In response to the reviewer's suggestion, we have revised Fig. 1b to include the geographical locations of the cities and places mentioned in the manuscript. This addition aims to enhance clarity and provide a better geographical context for the study area. We hope this revision improves the overall readability and effectiveness of the manuscript.

**Reviewer**: Lines 534-536 : we need wind vectors overlaid in the figure to understand these circulation aspects.

Reply: Thank you for your comment and thorough review. In response to the reviewer's suggestion, we have revised Figs. 7 and S2 and S3 by overlaying surface wind vectors on each

plot to better illustrate the circulation aspects discussed in the manuscript. We believe this addition enhances the clarity and understanding of the figures and improves the overall readability and effectiveness of the manuscript.

**Reviewer**: Line 566: there is not "trend" between two months, but a variation. We use the term "trend" for clear multiyear evolution with many time steps.

Reply: Thank you for your comment. In response to the reviewer's suggestion, we have revised and incorporated the analysis in the supplementary material. We hope this revision improves the precision and clarity of the manuscript.

**Reviewer**: Line 572 : too many digits are used to described the PBL height comparisons (e.g. 646.7 m ?) as compared to its precision.

Reply: In response to the reviewer's suggestion, we have revised the manuscript to remove excessive digits from the PBL height comparisons, ensuring that the values are presented without decimal places to reflect their precision better (please see the supplementary material). We believe this revision enhances the clarity and accuracy of the manuscript.

**Reviewer**: Line 655 : It should be clearly stated that the WRF-Chem model overestimates (positive bias) by a factor of 2 both background and peaks of NO2

Reply: We have revised the manuscript to clearly state that the WRF-Chem model overestimates, indicating a positive bias. This overestimation is observed when comparing the average values between the TROPOMI NO2 and the model over the UAE. This clarification has been incorporated into the revised manuscript (lines 584-587, Table 2).

**Reviewer**: Line 690 : these RMSE values are not compatible with the actual differences seen visually in the maps. If it is an unbiased RMSE that is calculated, the definition and name of the quantity should be revised.

Reply: Thank you for your comment and thorough review. Our analysis used the standard RMSE computation, not the unbiased RMSE. The RMSE values in our study represent the average error across the entire d03, which may not always correspond directly to the visual differences observed in the maps. The maps in Figs. 2, 3, and 4 show absolute differences, while the RMSE values in Table 2 and in the text summarize the overall model error for the domain. Thank you for highlighting this point.

**Reviewer**: How are the biases, RMSE, R of WRF-Chem in other regions in literature (East Asia and India) compare to those found in this study? Provide this comparison in percentage and the comparison with the results of the paper should be explicitly stated.

Reply: Thank you for your comment and thorough review. In response to the reviewer's suggestion, we have incorporated a comparison of the biases, RMSE, and R of WRF-Chem from other geographical regions, such as East Asia and India, as reported in the literature (lines 644-650). We believe this addition will enhance the clarity and comprehensiveness of the manuscript.

**Reviewer**: Ozone columns are often express in Dobson Units. For comparison, this unit should be used.

Reply: Thank you for your comment. In response to the reviewer's suggestion, we have revised the manuscript to express the ozone columns in Dobson Units (DU) for consistency and ease of comparison (lines 367-371, Table 2, Figure 4). We believe this modification will improve the clarity and comprehensiveness of the manuscript.

**Reviewer**: What is the influence of the altitude in the comparison between gaseous pollutants ? Where do averaging kernels peak? Model values without AVK should be shown as well to understand its influence.

Reply: Thank you for your comment. In response to the reviewer's suggestion, we have added a figure illustrating the averaging kernel matrix to show where the averaging kernels peak in the revised manuscript in the supplementary material as Figure S7. These revisions were made to improve the clarity and comprehensiveness of the manuscript. Due to the paper's length, we cannot include plots of model values without the averaging kernels. We hope the reviewer understands our decision.